# Role of Nrp1 in controlling cortical inter-hemispheric circuits

**Fernando Martín-Fernández[1], Ana Bermejo-Santos[2], Lorena Bragg-Gonzalo[1], Carlos G Briz[1], Esther Serrano-Saiz[2], Marta Nieto[1]***

[1]Department of Cellular and Molecular Biology, Centro Nacional de Biotecnología, Consejo Superior de Investigaciones Científicas (CNB-CSIC), Campus de Cantoblanco, Darwin, Madrid, Spain; [2]Centro de Biología Molecular Severo Ochoa (CSIC/UAM), Campus de Cantoblanco, Nicolás Cabrera, Madrid, Spain

**Abstract** Axons of the corpus callosum (CC) mediate the interhemispheric communication required for complex perception in mammals. In the somatosensory (SS) cortex, the CC exchanges inputs processed by the primary (S1) and secondary (S2) areas, which receive tactile and pain stimuli. During early postnatal life, a multistep process involving axonal navigation, growth, and refinement, leads to precise CC connectivity. This process is often affected in neurodevelopmental disorders such as autism and epilepsy. We herein show that in mice, expression of the axonal signaling receptor Neuropilin 1 (Nrp1) in SS layer (L) 2/3 is temporary and follows patterns that determine CC connectivity. At postnatal day 4, *Nrp1* expression is absent in the SS cortex while abundant in the motor area, creating a sharp border. During the following 3 weeks, *Nrp1* is transiently upregulated in subpopulations of SS L2/3 neurons, earlier and more abundantly in S2 than in S1. In vivo knock-down and overexpression experiments demonstrate that transient expression of *Nrp1* does not affect the initial development of callosal projections in S1 but is required for subsequent S2 innervation. Moreover, knocking-down *Nrp1* reduces the number of S2L2/3 callosal neurons due to excessive postnatal refinement. Thus, an exquisite temporal and spatial regulation of *Nrp1* expression determines SS interhemispheric maps.

**\*For correspondence:**
mnlopez@cnb.csic.es

**Competing interest:** The authors declare that no competing interests exist.

## Editor's evaluation

Your study highlights a novel role of Neuropilin 1 in regulating callosal connectivity at the level of the areal map with important insights on how areas mature and develop. The revisions of your manuscript have now clarified some of the methodological issue and we believe that it will be an important contribution to the field.

## Introduction

The cerebral cortex executes higher order functions by integrating information processed in different brain regions (*Hill and Walsh, 2005*). During evolution, the cortex of placental mammals expanded in size and functions. Together with this expansion, the brain acquired the corpus callosum (CC) for interhemispheric communication. The CC, the major axonal tract of the mammalian brain, is a tridimensional arrangement of myelinated axons forming networks with a precise hierarchical and topographical organization. Callosal axons branch and synapse only in certain contralateral locations and layers. They usually form columns in the border between cortical areas and outspread in layer (L) 2/3 and 5 of the mouse cortex (*Mitchell and Macklis, 2005*; *Courchet et al., 2013*; *Suárez et al., 2014b*; *Rodríguez-Tornos et al., 2016*; *Fenlon et al., 2017*). Thanks to the precise mapping of its connectivity, the CC allows many of our daily tasks, such as cognition, complex perception, or social

interactions. The CC is affected in many neurodevelopmental disorders such as autism spectrum disorders (ASD), epilepsy, schizophrenia, or bipolar disorders, due to the failure of mechanisms that we still do not understand (*Aboitiz and Montiel, 2003*; *Fame et al., 2011*; *Fenlon and Richards, 2015*).

In sensory perception, the CC allows complex responses by informing sensory areas of the sensory inputs received in the other hemisphere. In the somatosensory (SS) cortices, primary (S1) and secondary (S2) areas process first-order (tactile) and higher order (nociceptive) inputs, respectively (*Wise and Jones, 1976*; *Miller and Vogt, 1984*; *Rakic, 1988*; *Fenlon et al., 2017*; *De León Reyes et al., 2020*). Homotopic callosal connections communicate reciprocal sensory areas – they establish S1-S1 and S2-S2 connections. Heterotopic connections wire areas of a different order. Interhemispheric homotopic connectivity is favored and stronger. Callosal projections from S1 neurons branch profusely at the S1/S2 border but less at S2 and projections from S2 preferentially synapse in S2 (*Mitchell and Macklis, 2005*; *Courchet et al., 2013*; *Suárez et al., 2014a*; *Rodríguez-Tornos et al., 2016*; *Fenlon et al., 2017*). Other sensory systems replicate similar biased cortical interhemispheric connectivity between primary and secondary areas. Such maps are a consequence of developmental mechanisms that regulate each of the sequential steps of CC formation during a protracted period of embryonic and postnatal development. CC development initiates when cortical neurons project their axons to first traverse the cortical plate and then turn medially. Subsequently, axons cross the midline and reach the contralateral hemisphere following specific navigation signals. They then grow over the contralateral cortical plate and elaborate primitive columns in selected areas. These columnar bundles arborize profusely between the second and third postnatal week and are pruned in an activity-dependent manner within the third and fourth postnatal week. Within these periods, CC collaterals form additional columns in other territories (*Stanfield et al., 1982*; *Innocenti and Clarke, 1984*; *Dehay et al., 1986*; *Meissirel et al., 1991*; *Gibson and Ma, 2011*; *Innocenti, 2020*). Thus, exuberance and refinement are important contributors to the selection of proper connections, more so because virtually all neurons of the upper layers (L2/3 and L4) develop a transient callosal axon that crosses the midline (*De León Reyes et al., 2019*). These projections bear plasticity and have the potential to establish a mature CC connection in case of injury or developmental alterations, but most of them are eliminated between the second and fourth postnatal week by refinement. Only those neurons that succeed in synapsing with contralateral targets will become callosally projecting neurons (CPNs) of the mature cortex (*De León Reyes et al., 2019*). Our understanding of the molecular regulators of CC development is largely incomplete. We know some of the molecules that mediate early axonal navigation (*Hatanaka et al., 2009*; *Zhao et al., 2011*; *Zhou et al., 2013*) and that decisions to stabilize or refine callosal projections depend on the activity input from the distinct thalamic nuclei (*Mizuno et al., 2010*; *Suárez et al., 2014b*; *De León Reyes et al., 2019*). However, we lack major information on the intermediaries that determine the selection of contralateral targets.

Neuropilin-1 (Nrp1) is a receptor that mediates early steps of CC development through its binding to various ligands in association with signaling coreceptors (*Hatanaka et al., 2009*; *Zhao et al., 2011*; *Zhou et al., 2013*). In the embryo and early postnatal mouse, Nrp1 expression in the cortex follows a high to a low mediolateral gradient (*Zhao et al., 2011*; *Zhou et al., 2013*; *Muche et al., 2015*). At these stages, the interaction of Nrp1 with Semaphorin 3 (Sema3) C at the midline mediates axonal attraction and the crossing of the so-called pioneer callosal axons from neurons of the medial cortex. After midline crossing, the upregulation of the transmembrane protein EphrinB1 silences Nrp1/Sema3C signaling (*Gu et al., 2003*; *Niquille et al., 2009*; *Piper et al., 2009*; *Mire, 2018*). Once the CC pioneer path is created, Nrp1 plays an additional role in CC development by selecting the navigation routes of callosal projections from the motor and SS cortex thanks to its association with PlexinA1 and binding to Sema3A. This is possible because callosal axons from motor areas express high Nrp1 and low Sema3A levels while SS callosal axons display the inverse combination (*Takahashi et al., 1999*; *Fournier et al., 2000*; *Zhao et al., 2011*; *Wu et al., 2014*). The mutual repulsion imposed by gradients of Sema3A and Nrp1 determines early segregation of motor and SS projections into dorsal and ventral callosal routes, respectively, and contributes to lead callosal axons to their homotopic targets (*Zhou et al., 2013*). The subsequent temporal and spatial patterns of Nrp1 expression are poorly described and its possible functions unexplored. Herein, we investigated the expression and roles of Nrp1 during the development of CC circuits formed by L2/3 neurons of the somatosensory cortex. We found that *Nrp1* is transiently expressed in SSL2/3 subpopulations during the postnatal weeks that define the patterns of contralateral columnar connectivity, being earlier and more

frequently detected in S2 than in S1. Gain and loss-of-function experiments show that changing Nrp1 levels do not affect the initial formation of the S1/S2 column. However, both experimental conditions blocked two important steps of CC development occurring after the first postnatal week: exuberant arborization in S1/S2, and the formation of the S2 column. These experiments demonstrate that transient *Nrp1* expression determines the patterns of inter-areal callosal connectivity in SS.

## Results

### Nrp1 expression levels determine the pattern of SS contralateral innervation

Nrp1 shows a gradient both in mouse and human embryonic and early postnatal cortices (*Ren et al., 2006*; *Piper et al., 2009*). However, there are no detailed reports of its expression throughout postnatal life. To investigate the roles of Nrp1 in the formation of interhemispheric SS circuits, we characterized its expression using in situ hybridization (ISH) throughout representative postnatal stages of SS callosal development. At postnatal day (P) 4, when most SS callosal projections from L2/3 and L4 are beginning to cross the midline, we found that *Nrp1* mRNA expression is excluded from L2/3 and 4. Rather than a gradient, the absence of *Nrp1* in SS L2/3 and L4 neurons creates a sharp border between motor and SS cortices. The exclusion area coincides with the prospective SS barrel cortex (*Figure 1A–B*) identified by the presence of VGlut2$^+$ thalamic terminals (*Figure 1B*). More laterally, outside the SS cortex, L2/3 and L4 neurons tend to express low-to-intermediate levels of *Nrp1*. The area of *Nrp1* exclusion narrows caudally, coinciding with a smaller SS presumptive territory (*Figure 1A* and *Figure 1—figure supplement 1A*). Thus, the absence of *Nrp1* expression at P4 defines the nascent SS cortex. At P7, SS callosal axons begin the invasion of the contralateral cortical plate. VGlut2$^+$ terminals cluster in barrels in S1 but are diffusely distributed laterally, in S2. Expression of *Nrp1* is sparse in P7 S1L2/3 neurons and more abundant in S2L2/3 and S2L4 neurons (*Figure 1C* and *Figure 1—figure supplement 1B-C*). At P16, a stage in which callosal axons show collaterals and exuberant terminals that have not been yet pruned, *Nrp1* is found in individual cells uniformly scattered throughout S1 and S2 (*Figure 1D*). By contrast, most L2/3 neurons of the motor cortex have down-regulated *Nrp1*. These, together, eliminate differences in expression (*Figure 1—figure supplement 1D*). In P30 adult animals, most cells in the upper layers do not express *Nrp1* (*Figure 1E* and *Figure 1—figure supplement 1E*). Hence, this analysis revealed unreported developmental patterns of transient expression of *Nrp1*. Populations of L2/3 neurons in both S1 and S2 express *Nrp1* but with different temporal dynamics: expression follows a lateral to medial gradient within the SS and S2L2/3 neurons express *Nrp1* earlier and more abundantly.

### Knocking down and overexpressing Nrp1 in L2/3 neurons of the SS cortex reduces the S2 column

Next, we set to investigate the possible roles of the dynamic expression of Nrp1 in CC development by knocking down and overexpressing Nrp1. *Nrp1* null mutant mice are embryonically lethal (*Kitsukawa et al., 1997*). To bypass lethality, we performed in utero electroporation (IUE) of constructs knocking down Nrp1 (sh*Nrp1*) (*Figure 2A*). First, as a control for the shRNA efficiency, *Nrp1* levels were quantified in the cingulate cortex of P16 animals after electroporation. We observed a significant reduction in *Nrp1* expression in the electroporated area compared to the non-electroporated hemisphere (*Figure 2—figure supplement 1*). To analyze the effects on SS callosal projections, electroporations were performed at E15.5 to specifically target L2/3 neurons and with the electrodes oriented to S1 and S2. In parallel, we also analyzed the effects of overexpressing Nrp1 (CAG-Nrp1). Vectors were co-electroporated with a plasmid encoding GFP (CAG-GFP) for the characterization of electroporated neurons and their projections (*Figure 2B*). S1 barrel field area and the S2 area were distinguished by anatomical hallmarks such as the high density of L4 DAPI$^+$ nuclei that characterizes the barrels (*Paxinos and Franklin, 2004*). First, we examined the effects of our constructs on P30 animals. Brains electroporated with the control plasmid (CAG-GFP) showed that callosal projections from GFP$^+$ L2/3 neurons reproducibly elaborate separated axonal columns in the SS area of the contralateral hemisphere as described (*Courchet et al., 2013*; *Suárez et al., 2014a*; *Rodríguez-Tornos et al., 2016*). The main column – hereafter referred to as the S1/S2 column – locates at the border of the S1 and S2 area (*Figure 2A–C*, blue arrowheads). Another less dense but very similar axonal column appears in

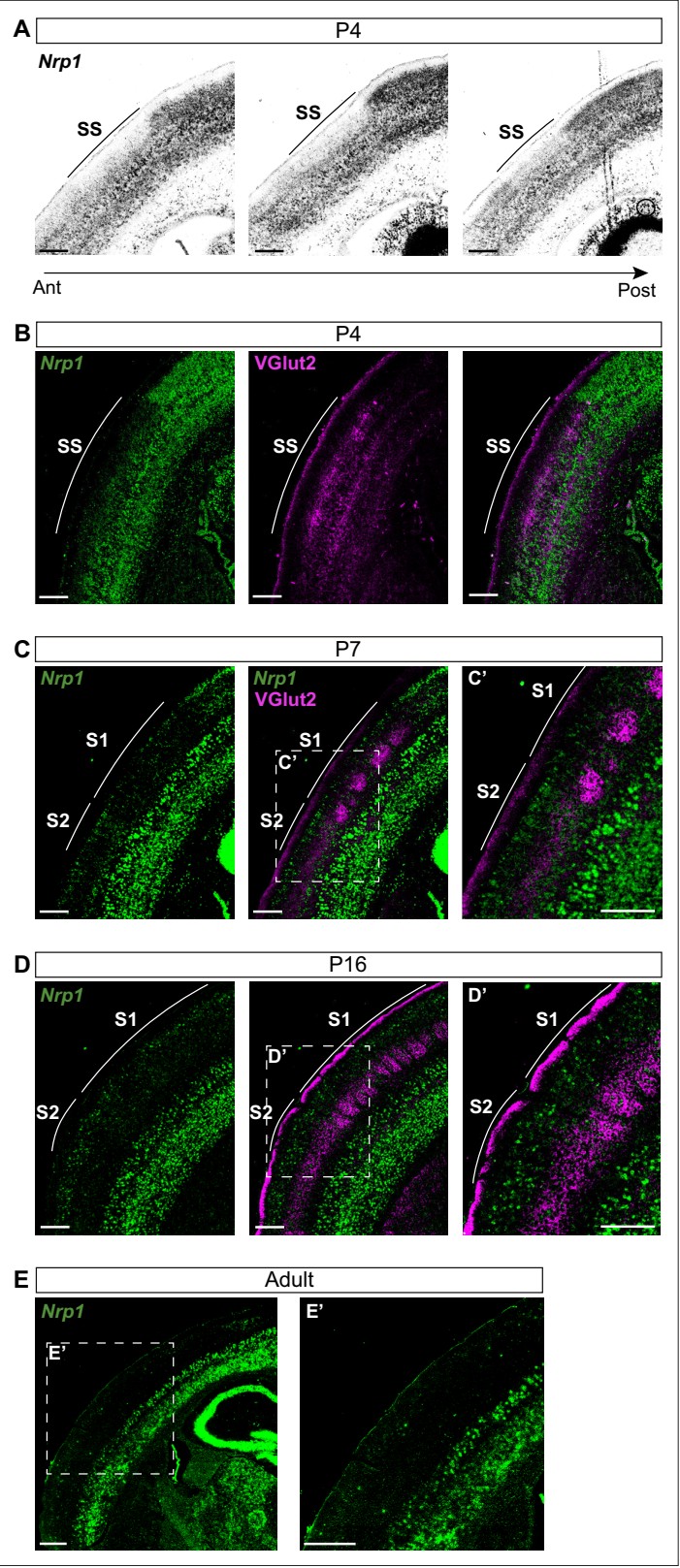

**Figure 1.** *Nrp1* expression in somatosensory cortex during postnatal development. (**A**) In situ hybridization (ISH) analysis of *Nrp1* expression in coronal sections of P4 brain. (**B–D**) ISH in combination with fluorescent antibody staining of VGlut2. Green = *Nrp1*, Magenta = VGlut2. Scale bar = 300 µm. (**B**) Analysis of P4 brains. VGlut2 signal is located in the somatosensory (SS) area. (**C**) Sections of P7 brains. VGlut2 expression delimitates the barrel field

*Figure 1 continued on next page*

*Figure 1 continued*

area in S1. S2 is located lateral to the barrel field. (**D**) At P16, expression differences of *Nrp1* in S1 and S2L2/3 neurons disappeared. (**E**) *Nrp1* expression in brain sections of adult mice. The upper layers of the cortex lose *Nrp1* expression in the SS cortex. Green = *Nrp1*. Scale bar = 500 μm.

The online version of this article includes the following figure supplement(s) for figure 1:

**Figure supplement 1.** *Nrp1* expression in the postnatal somatosensory cortex.

the lateral border of S2 henceforth called the S2 column (*Figure 2A–C*, magenta arrowheads). Within the S1/S2 column, axons branch more profusely in L2/3 and L5 while within the S2 column they branch mainly at L2/3 (*Figure 2C*). Knocking down or overexpressing Nrp1 in L2/3 neurons did not alter midline crossing nor block the axonal invasion of contralateral territories. Both conditions produced no apparent phenotype in the S1/S2 column but visibly reduced axons in S2 (*Figure 2D–E*). To quantify these phenotypes, we measured the GFP fluorescence pixels in contralateral regions of interest (ROIs) delineating SS areas or columns. To account for differences in electroporation efficiency, we normalized these values of axonal occupancy to the number of GFP$^+$ neurons in the ipsilateral hemisphere (see Materials and methods) (*Rodríguez-Tornos et al., 2016*; *Briz et al., 2017*). First, the analysis of the total contralateral innervation showed average values and dispersion indistinguishable in controls, sh*Nrp1*, and CAG-Nrp1 conditions (*Figure 2F*), and tendencies consistent with reductions only in specific areas. We, therefore, quantified separately the GFP$^+$ signal in the S1/S2 column (*Figure 2G*) and the S2 column (*Figure 2H*). We observed a reduction in GFP$^+$ signal in S2 columns in both sh*Nrp1* and CAG-Nrp1 electroporated brains, greater in the overexpressing condition (*Figure 2H*). These experiments indicated that altering Nrp1 levels does not cause an overall impairment of innervation but modifies the patterns of innervation. Alternative methods of normalization – to the ipsilateral whole area's fluorescence or to other contralateral areas – rendered equivalent results (see Materials and methods and *Figure 2—figure supplement 2*). Differences were not due to neuronal death or electroporation efficiency because the number and distribution of electroporated GFP$^+$ neurons were equivalent across conditions (*Figure 2—figure supplement 3*). Thus, increasing or reducing Nrp1 levels in SSL2/3 neurons reduces callosal axons in the contralateral S2 area of mature brains. This indicates that transient expression of Nrp1 is required to innervate S2 areas.

## Nrp1 levels orchestrate S2 homotopic callosal innervation

We then evaluated topographic changes in the origin of projections to S1 or S2 in the different electroporating conditions. Using stereotaxic coordinates, we performed classic axonal retrograde tracing by injecting fluorescent conjugates of the cholera toxin subunit B (CTB-555) in the cortical plate of the non-electroporated hemisphere. This procedure labels the subset of neurons projecting to the site of injection and identifies the location of their soma in the S1 or S2 areas of the opposite hemisphere (*Figure 3A*). We injected P28 animals either in S1, close to the S1/S2 column (S1/S2 injections) (*Figure 3A–B* and *Figure 3—figure supplement 1A-J*) or in the S2 column (*Figure 3A and C* and, *Figure 3—figure supplement 1K-U*) and analyzed the distribution of the GFP$^+$CTB$^+$ L2/3 CPNs at P30 (*Figure 3A*). As a retrospective control of the injection site, we confirmed that, in addition to cortical neurons, our S1/S2 injections preferentially labeled thalamic neurons of the ventral postero-medial nuclei (VPM) (*Figure 3—figure supplement 2A-C*), while our injections in the S2 column labeled neurons of the posterior nucleus (Po) (*Figure 3—figure supplement 2D-F*) (see Materials and methods). For each type of injection, after counting the double-positive CPNs, we assessed their relative distribution in S1 and S2. For injections in the S1/S2 column, we calculated the ratio of GFP$^+$CTB$^+$ neurons in S1 vs. the number in S2 (homotopic projections vs. heterotopic projections). This analysis showed that in controls most axons that form the S1/S2 column are homotopic projections from S1 since GFP$^+$ S1L2/3 neurons were labeled 1.5 times more frequently than those in S2 (*Figure 3D and J*). With these injections, we detected no changes in the composition of axons forming the contralateral S1 columns of sh*Nrp1* or CAG-Nrp1 brains, although both conditions showed a tendency to smaller relative contributions of S1 projections (*Figure 3E–F and J*). Hence, since the preferential innervation of S1/S2 by homotopic S1 projections is unaffected by our manipulations of Nrp1 expression, this indicated that this selectivity does not depend on the transient expression of Nrp1. For animals injected in the S2 column, we calculated the ratio of GFP$^+$CTB$^+$ neurons found in S2 (homotopic projections) vs. those labeled in S1 (heterotopic) (*Figure 3G–I and K*). In the control condition, homotopic S2L2/3

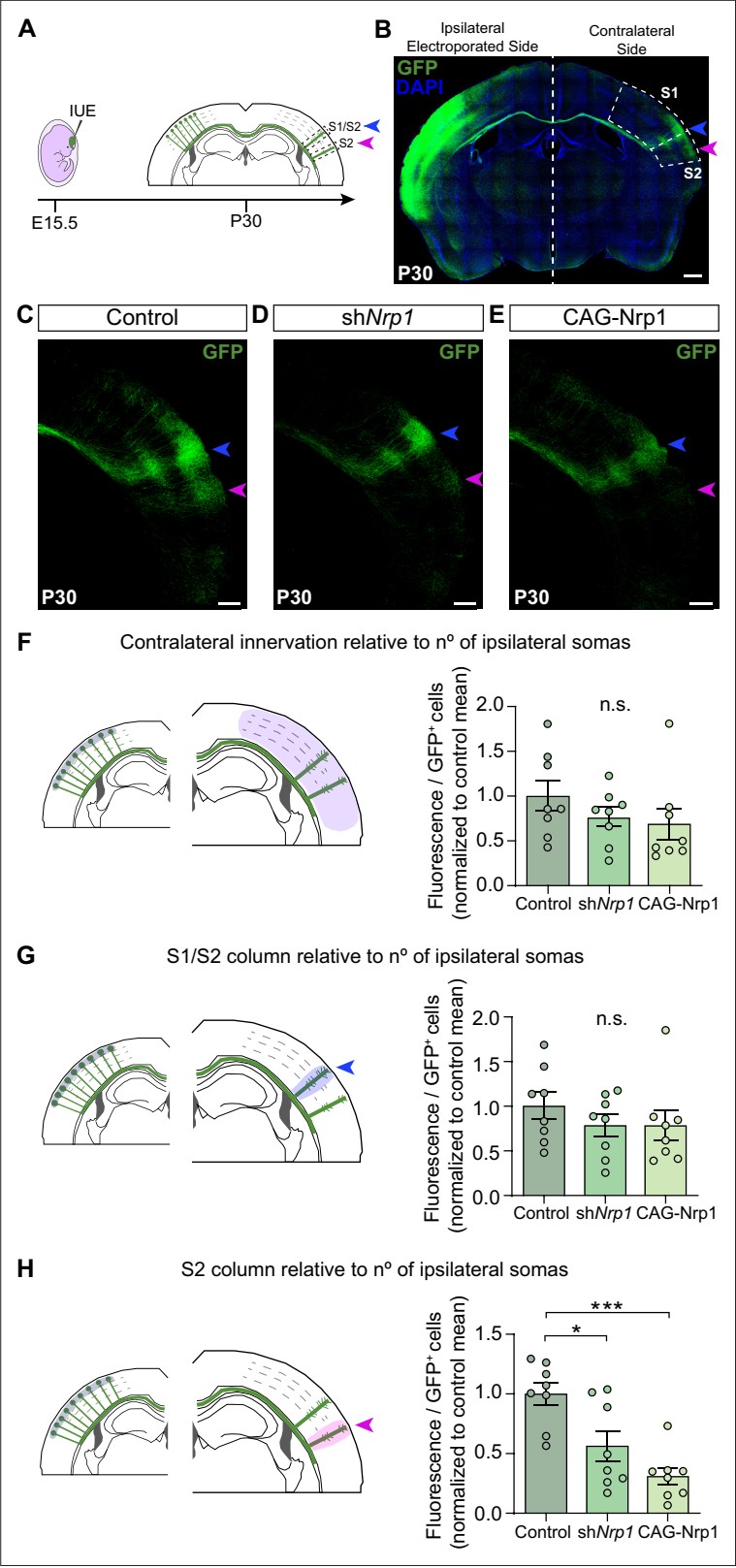

**Figure 2.** Distribution of GFP⁺ callosal axons in the contralateral hemisphere after knocking down or over-expressing Nrp1. (**A**) Scheme of the experimental approach. Callosal projections from electroporated L2/3 neurons establish the S1/S2 axonal column (blue arrow) and the S2 column (magenta arrow) in the non-electroporated hemisphere. (**B**) Confocal image of a coronal section of P30 control brain IUE at E15.5 with CAG-GFP. Dashed

*Figure 2 continued*

boxes indicate the divisions into the primary (S1) and secondary (S2) somatosensory cortex. + = GFP, Blue = DAPI. Scale bar = 500 µm. (**C–E**) High magnification images showing the contralateral hemisphere of P30 brains IUE with control plasmid (**C**), sh*Nrp1* (**D**), or CAG-Nrp1 (**E**). GFP+ axons (green), S1/S2 (blue arrow), S2 columns (magenta arrow). Scale bar = 300 µm. (**F–H**) Quantifications of axonal distribution in the contralateral hemisphere. The left panels depict schemes showing the selected ROIs in which GFP+ is quantified (shaded areas). Graphs show values of GFP signal relative to the number of L2/3GFP+ neurons quantified in the opposite (ipsilateral) electroporated hemisphere of the same coronal section. Innervation values are normalized to the mean value of controls. Mean ± SEM (n = 8 mice, 2 sections per brain, in all conditions). S1/S2 column (blue arrow), S2 column (magenta arrow). Statistics (n total = 24): (**F**) One-way ANOVA: p-value = 0.3044 (n.s.). (**G**) One-way ANOVA: p-value = 0.4762 (n.s.). (**H**) One-way ANOVA: p-value = 0.0003. Posthoc with Tukey's test: * p-value $_{Control – shNrp1}$ = 0.0157, *** p-value $_{Control – CAG-Nrp1}$ = 0.0002. Source data are provided as a Source Data file.

The online version of this article includes the following source data and figure supplement(s) for figure 2:

**Source data 1.** Raw data of measurements.

**Figure supplement 1.** The electroporation of sh*Nrp1* plasmid at E15.5 reduces the expression of *Nrp1* transcripts in P16 brains.

**Figure supplement 1—source data 1.** Raw data of qPCR.

**Figure supplement 2.** Analysis of contralateral innervation of SS cortex at P30 upon Nrp1 modifications.

**Figure supplement 2—source data 1.** Raw data of measurements.

**Figure supplement 3.** Quantification of GFP+ neurons in the electroporated hemisphere.

**Figure supplement 3—source data 1.** Raw data-countings.

---

projections are the main contributors to the GFP+ S2 column (nearly 2.5 ratio) (*Figure 3G and K*). Thus, in controls, there is higher homotopic selectivity within S2 connections as compared to S1 maps. Both knocking down or overexpressing Nrp1 decreased the proportions of S2L2/3 CPNs labeled with CTB from injections in the contralateral S2 column. They reduced the value of homotopic/heterotopic innervation drastically (1.5 ratios) (*Figure 3H–I and K*). These data confirm the reduction of the GFP+ S2 column in both sh*Nrp1* and CAG-Nrp1 conditions (*Figure 2D, E and H*) and indicate they are due to a greater loss of homotopic S2L2/3 projections compared to the loss of heterotopic S1L2/3 axons. Notably, the diminished S2 columns and the preserved S1/S2 columns are all formed by similar proportions of S1 and S2L2/3 projections. This suggests that upon equal levels of Nrp1 expression, S2L2/3 axons lose their advantage for homotopic innervation. These shifts in CTB+ CPNs distributions were not caused by differences in labeling efficiency, as we detected no changes in the number of non-electroporated CTB+ cells among conditions (*Figure 3—figure supplement 3*). Together, the data demonstrated that knocking down or overexpressing Nrp1 impair the growth in the contralateral S2 areas of callosal axons from S1L2/3 neurons but affect more the homotopic projections from S2L2/3 neurons. By contrast, manipulating Nrp1 levels does not affect homotopic innervation of S1 in the same way.

## Changes in Nrp1 expression alter developmental growth and refinement of callosal projections

Next, we investigated if changes in developmental axonal dynamics lead to the distinct topography of callosal connectivity in P30 sh*Nrp1* and CAG-Nrp1 brains. To this end, we analyzed and compared axonal distributions of P10, P16, and P30 animals electroporated at E15.5 (*Figure 4A*). The laminar and area distributions of the electroporated cells were equivalent in all conditions (*Figure 4—figure supplement 1*). At P10, the analysis revealed the S1/S2 column and the absence of the S2 column indistinguishable in all conditions. We detected some axons in S2 but they showed minimal branching as if initiating invasion (*Figure 4B–D*). This showed that the formation of the S1/S2 column precedes the development of the contralateral S2 branches. Quantifications of the GFP+ projections forming the S1/S2 column and of axons in S2 demonstrated no significant differences between P10 control, sh*Nrp1*, and CAG-Nrp1 brains (*Figure 4H–K*, and *Figure 4—figure supplement 2*). Thus, changing Nrp1 levels does not affect the early invasion of the contralateral cortex nor the first establishment of an axonal column at the S1/S2 border. At P16, as development proceeds, the analysis of callosal axons in controls indicated major growth in S1 and more in S2, compared to P10. In the

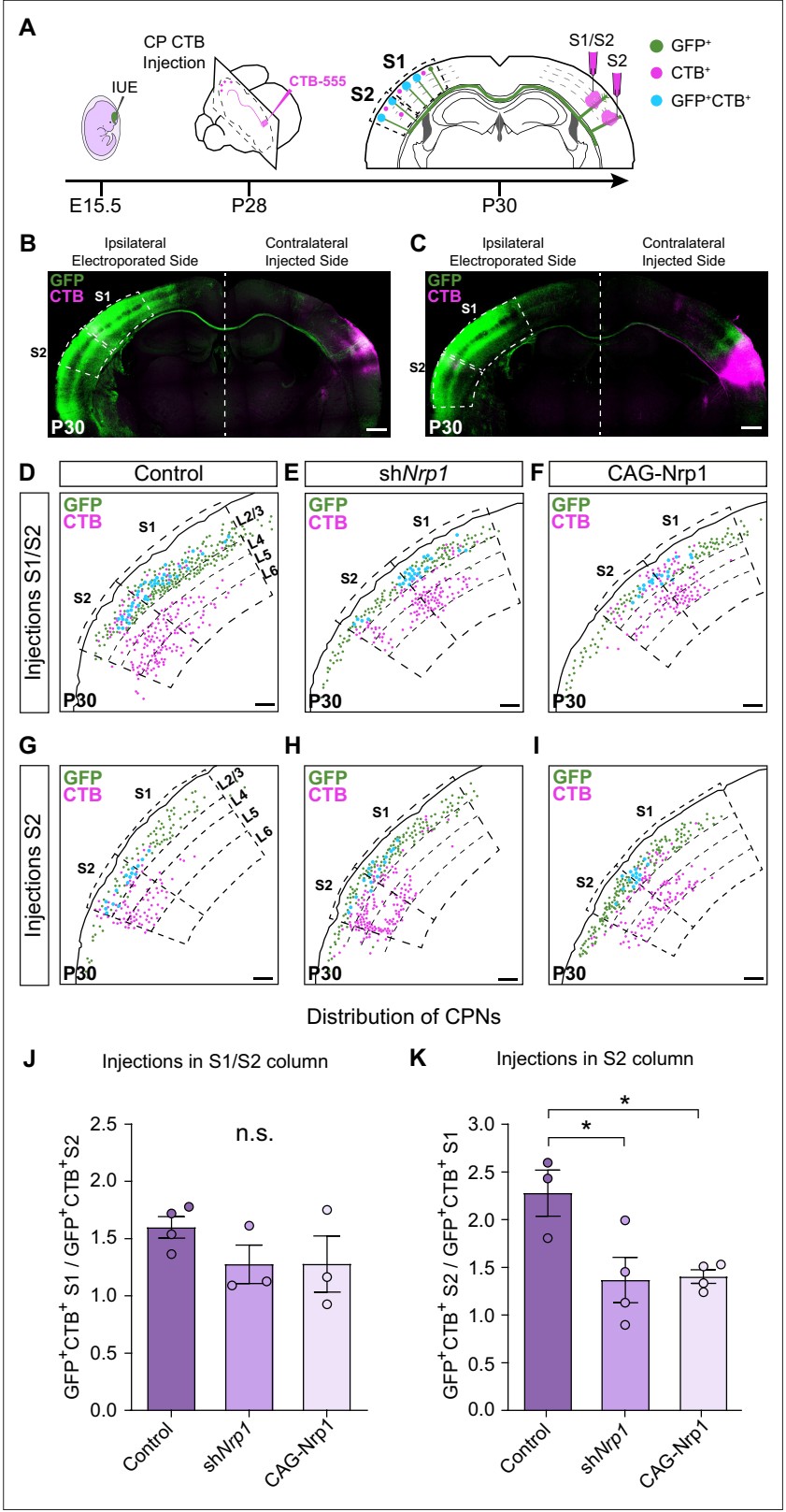

**Figure 3.** Analysis of homotopic and heterotopic projections in control, sh*Nrp1*, and CAG-Nrp1 IUE brains. (**A**) Experimental workflow. After IUE at E15.5, brains are stereotaxically injected with CTB in the cortical plate (CP). Separate animals are injected in the S1/S2 column or the S2 column at P28 and 2 days after (**P30**) the numbers of GFP+CTB+ CPNs are quantified in the S1 and S2 areas of the electroporated hemisphere. (**B–C**) Tilescan images

*Figure 3 continued on next page*

*Figure 3 continued*

of control IUE brains injected in S1/S2 (**B**) or S2 (**C**). Green = GFP, Magenta = CTB. Scale bar = 500 μm. (**D–I**) Representative examples of the analysis reporting the location of GFP⁺ (green dots), CTB⁺ (magenta dots), and GFP⁺CTB⁺ (blue dots) neurons in injected brains as in (**A**). Scale bar = 300 μm. (**J**) Quantification of the distribution of GFP⁺CTB⁺ cells in brains injected in the S1/S2 column. The values represent the number of GFP⁺CTB⁺ cells in S1 divided by the number of GFP⁺CTB⁺ cells in S2 in the same section. Mean ± SEM (n ≥ 3 mice, ≥ 2 sections per brain, in all conditions). One-way ANOVA (n total = 10): p-value = 0.3155 (n.s.). (**K**) Quantification of the distribution of GFP⁺CTB⁺ cells in brains injected in S2. The values represent the ratio of the number of GFP⁺CTB⁺ in S2 divided by the number of GFP⁺CTB⁺ cells in S1 in the same section. Mean ± SEM (n ≥ 3 mice, ≥ 2 sections per brain, in all conditions). One-way ANOVA (n total = 11): p-value = 0.0218. Posthoc with Tukey's test: * p-value $_{Control - shNrp1}$ = 0.0288; * p-value $_{Control - CAG-Nrp1}$ = 0.0346. Source data are provided as a Source Data file.

The online version of this article includes the following source data and figure supplement(s) for figure 3:

**Source data 1.** Raw data-countings.

**Figure supplement 1.** Images of the injection sites in stereotaxic surgeries.

**Figure supplement 2.** The whole image of a coronal section-including the thalamus-for the retrospective control of stereotaxic injections.

**Figure supplement 3.** Analysis of the location of CPNs in the somatosensory cortex of electroporated brains.

**Figure supplement 3—source data 1.** Raw data-countings.

contralateral hemisphere, GFP⁺ axons establish S1/S2 and S2 columns very similar to those of P30 animals (*Figure 4E and H–K*). Thus, the S2 column develops due to growth within the P10-P16 time window. This growth was significantly altered in sh*Nrp1* or CAG-Nrp1 electroporated brains, which showed values at P16 indistinguishable from those at P10, both in S1 and S2, and diminished when compared to the P16 control (*Figure 4H–K*, and *Figure 4—figure supplement 2*). These reductions did not correlate with shifts of dorsoventral navigation routes. There were no significant differences in the dorsoventral distribution of GFP signal at the CC midline in control, sh*Nrp1*, and CAG-Nrp1 electroporated brains. In all, GFP⁺ axons crossed by the most ventral two-thirds of the CC in indistinguishable manners (*Figure 4—figure supplement 3*). Thus, the comparison of postnatal stages indicates that GFP⁺ callosal axons of sh*Nrp1* and CAG-Nrp1 electroporated L2/3 neurons stall their development after P10. The effects caused by this developmental stagnation seem ameliorated at P30 mainly because in controls contralateral branches decrease from P16 to P30 (*Figure 4H–J*) as a consequence of the pruning of exuberant arbors (*O'Leary, 1992*; *De León Reyes et al., 2019*). The reductions in S2 showed no recovery at P30. Overall, these experiments show that altering the dynamic regulation of Nrp1 expression blocks the developmental progression of axons after P10, thereby impeding the formation of an S2 column and the growth and refinement of exuberant arbors in the S1/S2 column.

## Knocking down Nrp1 eliminates populations of S2L2/3 CPNs by refinement

The decreases in contralateral GFP⁺ axons could be due to reduced axonal branching from unaltered numbers of CPNs or to reductions in CPN numbers. We recently showed that as part of their normal differentiation, most L2/3 neurons develop axons that project callosally and are then eliminated by area-specific activity-dependent mechanisms. This refinement generates the two majors L2/3 mature subpopulations: ipsilateral-only L2/3 projecting neurons and L2/3 CPNs. Such process occurs during a protracted period of postnatal development ending around P30, but is more intense during the first two weeks of life (P1-P16), coinciding with changes in Nrp1 expression. At P16, the proportion of S1L2/3 CPNs is very similar to that of the adult, while S2L2/3 populations still undergo some CPN refinement between P16-P30 (*De León Reyes et al., 2019*). We next investigated a possible influence in CPN refinement the reductions in S2 innervation caused by our manipulations of Nrp1 expression. To this end, we analyzed CPN numbers in the SS cortex of control, sh*Nrp1*, and CAG-Nrp1 electroporated brains at P16 and P30. For this, instead of targeting the cortical plate, we injected CTB-555 directly in the CC in the non-electroporated hemisphere (*Figure 5A–C*). This procedure labels all neurons with an axon crossing the midline, including those in the process of developing or refining their callosal projections (*De León Reyes et al., 2019*). In controls, quantifications showed proportions of S1L2/3 and S2L2/3 CPNs undistinguishable to those previously reported, indicating that IUE does

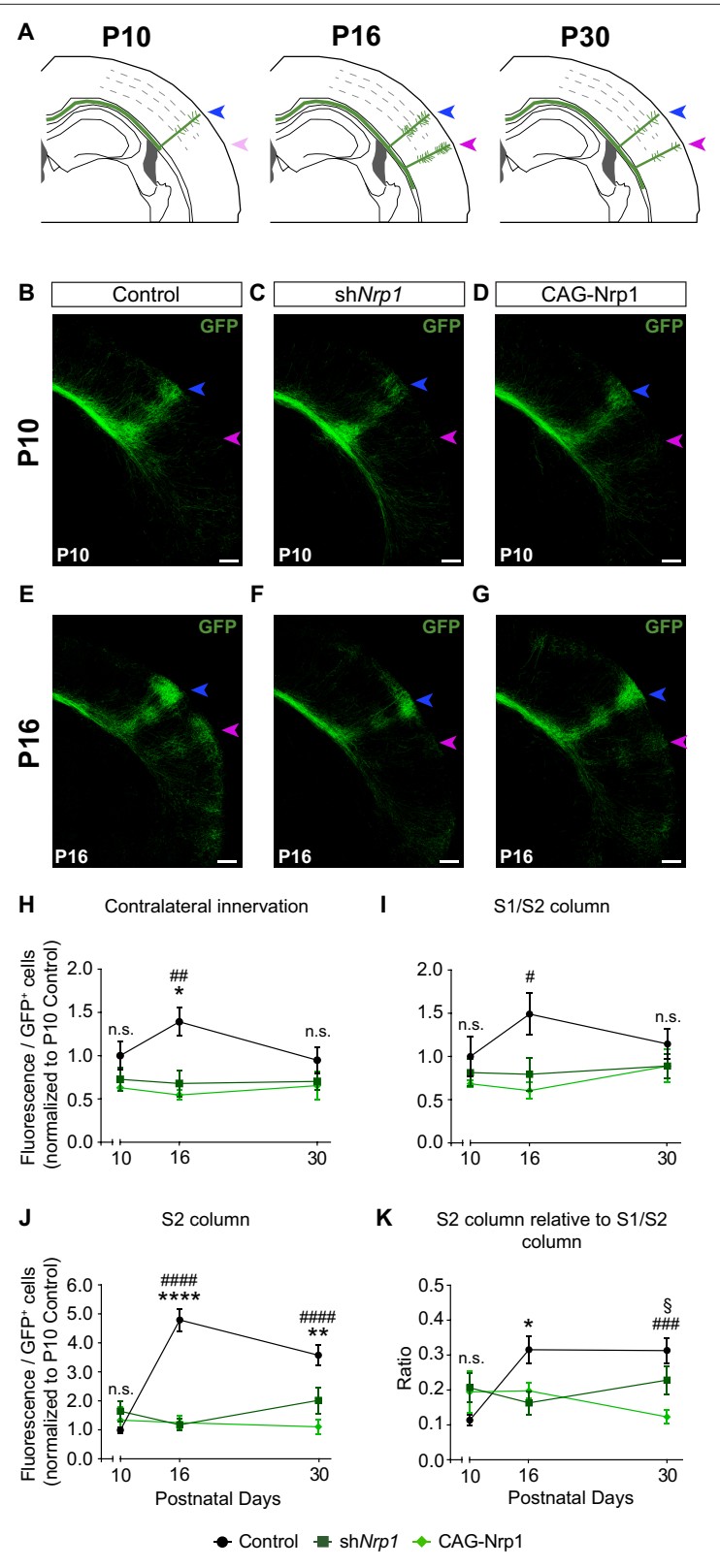

**Figure 4.** Comparisons of the postnatal changes of contralateral axons during the P10 to P30 window upon manipulations in Nrp1 expression. (**A**) Schematic representation of contralateral innervation dynamics during postnatal development. (**B–G**) Tilescan images of the contralateral hemisphere of IUE brains analyzed at P10 and P16. Blue arrow = S1/S2 column. Magenta arrow = S2 column. Green = GFP. Scale bar = 300 μm. (**H–K**)

*Figure 4 continued on next page*

*Figure 4 continued*

Quantifications of GFP+ innervation in the indicated area. GFP values are expressed relative to the number of L2/3 GFP+ neurons in the electroporated hemisphere and normalized to the mean value of P10 control. Mean ± SEM (n ≥ 3 mice, 2 sections per brain, in all conditions). Statistics (n total = 47): (**H**) Two-way ANOVA: p-value $_{Dynamics\ of\ contralateral\ innervation}$ = 0.3938; p-value $_{Postnatal\ day}$ = 0.6903; p-value $_{Experimental\ condition}$ = 0.0010. Posthoc with Tukey's test: * p-value $_{Control\ P16\ –\ shNrp1\ P16}$ = 0.0156; ## p-value $_{Control\ P16\ –\ CAG-Nrp1\ P16}$ = 0.0037. (**I**) Two-way ANOVA: p-value $_{Dynamics\ of\ S1/S2\ column}$ = 0.4979; p-value $_{Postnatal\ day}$ = 0.6520; p-value $_{Experimental\ condition}$ = 0.0125. Posthoc with Tukey's test: # p-value $_{Control\ P16\ –\ CAG-Nrp1\ P16}$ = 0.0157. (**J**) Two-way ANOVA: p-value $_{Dynamics\ of\ S2\ column}$ <0.0001; p-value $_{Postnatal\ day}$ = 0.0078; p-value $_{Experimental\ condition}$ <0.0001. Posthoc with Tukey's test: **** p-value $_{Control\ P16\ –\ shNrp1\ P16}$ <0.0001; #### p-value $_{Control\ P16\ –\ CAG-Nrp1\ P16}$ <0.0001; ** p-value $_{Control\ P30\ –\ shNrp1\ P30}$ = 0.0022; #### p-value $_{Control\ P30\ –\ CAG-Nrp1\ P30}$ <0.0001. (**K**) Two-way ANOVA: p-value $_{Dynamics\ S2\ column\ relative\ to\ S1/S2\ column}$ = 0.0057; p-value $_{Postnatal\ day}$ = 0.2288; p-value $_{Experimental\ condition}$ = 0.0737. Posthoc with Tukey's test: * p-value $_{Control\ P16\ –\ shNrp1\ P16}$ = 0.0392; ### p-value $_{Control\ P30\ –\ CAG-Nrp1\ P30}$ = 0.0002; § p-value $_{shNrp1\ P30\ –\ CAG-Nrp1\ P30}$ = 0.0446. Data for P30 are from *Figure 2* and *Figure 2—figure supplement 2*. Source data are provided as a Source Data file.

The online version of this article includes the following source data and figure supplement(s) for figure 4:

**Source data 1.** Raw data of measurements.

**Figure supplement 1.** Quantification of GFP+ neurons in the electroporated hemisphere and rostro-caudal classification of analyzed sections.

**Figure supplement 1—source data 1.** Raw data-countings and sterotaxic coordinates.

**Figure supplement 2.** Analysis of contralateral innervation of SS cortex at P10 and P16 upon Nrp1 modifications.

**Figure supplement 2—source data 1.** Source data file for *Figure 4—figure supplement 2*.

**Figure supplement 3.** Analysis of the dorsoventral distribution of axons at the midline.

**Figure supplement 3—source data 1.** Raw data of measurements.

---

not affect CPN development (*Figure 5D–E and G–H* and, *Figure 5—figure supplements 1–2*; *Fame et al., 2011*; *De León Reyes et al., 2019*). The number of P16 or P30 CPNs was not modified upon overexpression of Nrp1 (*Figure 5D and G*). However, CPN numbers were altered in sh*Nrp1* brains (*Figure 5D–I*). In P16 control brains, 50% of GFP+ S1L2/3 neurons were CTB+ (*Figure 5D and E*), while this number increased up to 65% in sh*Nrp1* brains (*Figure 5D and F*). At P30, the final number of GFP+ S1L2/3 CPNs in sh*Nrp1* and control electroporated brains were indistinguishable (*Figure 5D*). Thus, since we observed no evidence of neuronal death in sh*Nrp1* electroporated neurons (*Figure 2—figure supplement 3*), late postnatal refinement normalizes transient increases of CPNs induced by knocking down Nrp1. By contrast, while we detected no changes in the number of GFP+ S2L2/3 CPNs in sh*Nrp1* targeted brains at P16 (*Figure 5G*), they were reduced at P30 (*Figure 5G–I*). Thus, these experiments demonstrated that the numbers of L2/3 CPNs in sh*Nrp1* or CAG-Nrp1 electroporated P16 brains are equal or higher than in controls in all areas. Hence, the reductions in GFP+ branches we observed are the result of scarce arborization in the contralateral cortical plate and not due to decreases in CPNs. This again supports stalled axonal maturation in both sh*Nrp1* and CAGNrp1 conditions. In sh*Nrp1* electroporated brains, this progresses to increased rates of elimination of GFP+ S2L2/3 callosal axons, possibly due to the refinement of axons without terminal synapses neither in S1/S2 nor in S2. These data demonstrate that by regulating terminal axonal callosal maturation, transient Nrp1 expression determines S2 innervation and the number of S2 homotopic CPNs.

## Discussion

We herein demonstrate that Nrp1 functions regulate the postnatal development of SS callosal circuits. Gain- and loss-of-function experiments demonstrate that transient expression of Nrp1 promotes the elaboration of exuberant axons and is required for the establishment of projections in S2. Both these processes occur after midline crossing during mid and late stages of postnatal development – the second and third postnatal weeks in mice. Because S1 and S2L2/3 neurons express *Nrp1* with distinct temporal and spatial patterns, post-crossing functions of Nrp1 contribute to a hierarchical organization of bilateral somatosensory circuits.

Previous studies have shown that differences in *Nrp1* mRNA levels at P0 determine an orderly organization of motor and SS callosal axons (*Zhou et al., 2013*). However, the patterns of *Nrp1* expression at later stages were not described. We performed a detailed analysis of the expression

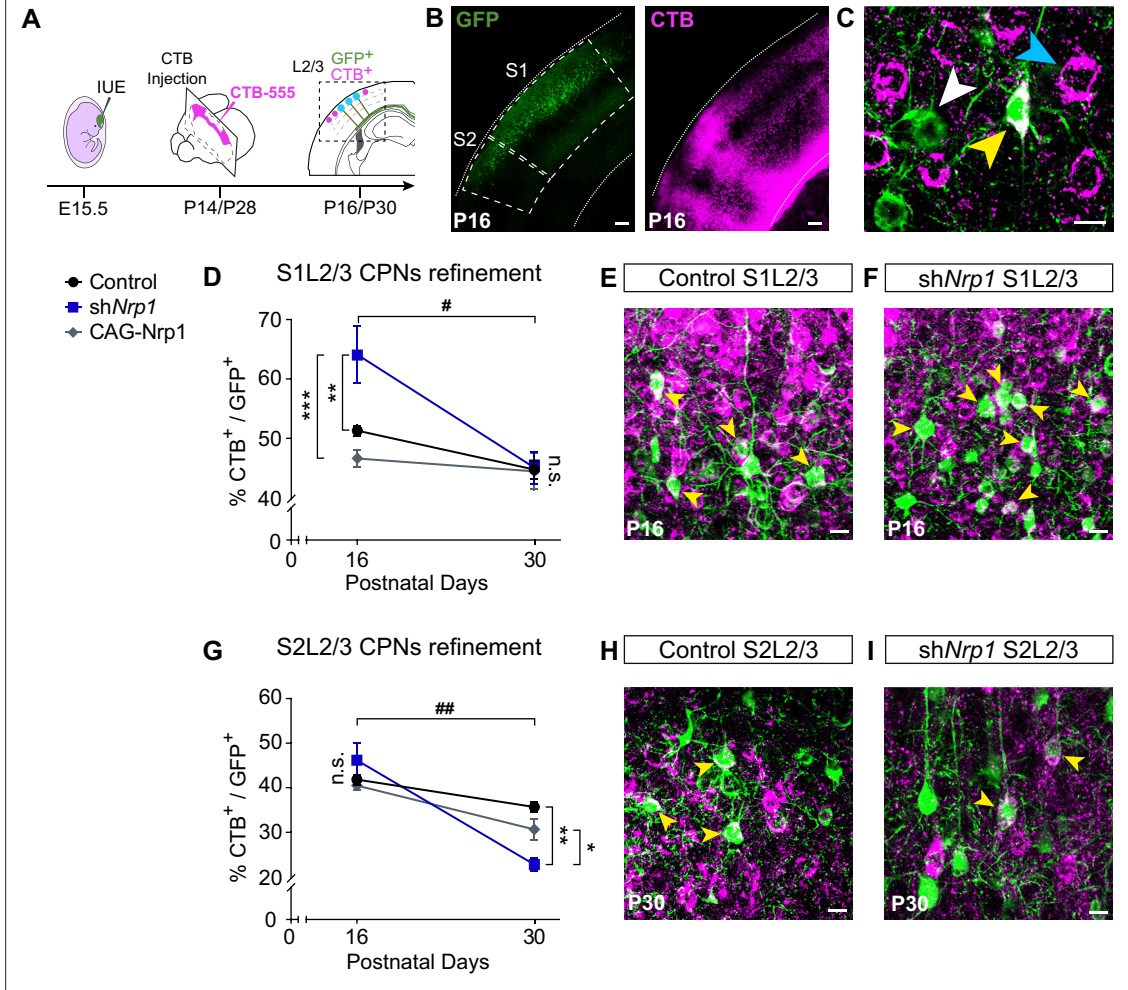

**Figure 5.** CPNs refinement during the P16 to P30 developmental window. (**A**) Scheme of the experimental workflow. To analyze the effect of developmental refinement on the number of CPNs in the electroporated population, stereotaxic CTB injections at the midline were performed, after IUE at E15.5. (**B**) Images showing ipsilateral cortices of electroporated P16 brains with S1 and S2 regions delimited by dashed lines. CTB signal is found in axonal columns and somas. Green = GFP. Magenta = CTB. Scale bar = 300 µm. (**C**) High-magnification image of L2/3 neurons in an injected P16 brain. White arrowhead = GFP⁺ neurons, blue arrowhead = CTB⁺ neurons, yellow arrowhead = GFP⁺CTB⁺ neurons. Scale bar = 10 µm. (**D**) Proportion of GFP⁺ CPNs (number of GFP⁺CTB⁺/number of GFP⁺) in S1 area in P16 and P30 brains. Mean ± SEM (n ≥ 3 mice, 2 sections per brain, in all conditions). Two-way ANOVA (n total = 21): ## p-value $_{S1L2/3\ CPNs\ refinement}$ = 0.0120; p-value $_{Experimental\ condition}$ = 0.0075; p-value $_{Postnatal\ day}$ = 0.0003. Posthoc with Tukey's test: ** p-value $_{Control\ P16\ -\ shNrp1\ P16}$ = 0.0064; *** p-value $_{shNrp1\ P16\ -\ CAG-Nrp1\ P16}$ = 0.0008. (**E–F**) Merge images of control (**E**) and sh*Nrp1* (**F**) S1L2/3 neurons at P16. Yellow arrowheads = GFP⁺CTB⁺ neurons. Scale bar = 10 µm. (**G**) Quantifications of CPNs in S2 at P16 and P30. Mean ± SEM (n ≥ 3 mice, 2 sections per brain, all conditions). Two-way ANOVA (n total = 21): ## p-value $_{S2L2/3\ CPNs\ refinement}$ = 0.0029; p-value $_{Experimental\ condition}$ = 0.1358; p-value $_{Postnatal\ day}$ <0.0001. Posthoc with Tukey's test: ** p-value $_{Control\ P30\ -\ shNrp1\ P30}$ = 0.0021; * p-value $_{shNrp1\ P30\ -\ CAG-Nrp1\ P30}$ = 0.0448. (**H–I**) Merge images of control (**H**) and sh*Nrp1* (**I**) S2L2/3 neurons at P16. Yellow arrowheads = GFP⁺CTB⁺. Scale bar = 10 µm. Source data are provided as a Source Data file.

The online version of this article includes the following source data and figure supplement(s) for figure 5:

**Source data 1.** Source data file for *Figure 5*.

**Figure supplement 1.** Proportions of non-electroporated and electroporated CPNs at P16.

**Figure supplement 1—source data 1.** Raw data-countings.

**Figure supplement 2.** Proportions of non-electroporated and electroporated CPNs at P30.

**Figure supplement 2—source data 1.** Raw data-countings.

of *Nrp1* transcripts in the postnatal SS cortex at representative stages of CC development. Unfortunately, we could not assess the expression of Nrp1 protein. As in previous studies, our antibody staining detected the protein only in midline axons (*Piper et al., 2009*; *Zhao et al., 2011*; *Zhou et al., 2013*; *Lim et al., 2015*). Using ISH, we found that *Nrp1* expression is excluded from SSL2/3 neurons

 Research article

at early postnatal stages. During the next postnatal weeks, *Nrp1* is detected in scattered L2/3 cells, more abundantly and first in S2L2/3, and after in S1L2/3. In mature SSL2/3 neurons, *Nrp1* is down-regulated. The data indicates more persistent expression in S2L2/3 neurons, however, from the ISH, we cannot distinguish if all SSL2/3 neurons express *Nrp1* transiently or only restricted subpopulations of L2/3 neurons eventually activate its expression. Besides, the early absence of *Nrp1* coincides with the prospective S1. Since the S1 and S2 areas receive different input from first-order and higher order thalamic nuclei (*Inan and Crair, 2007*; *Pouchelon et al., 2014*), this suggests that S1 input from thalamic afferents might reduce *Nrp1* expression in S1L2/3 neurons.

Our findings indicate that transient upregulation of Nrp1 is necessary to promote the developmental progression of callosal innervation after P10, therefore, both overexpressing and knocking-down Nrp1 have similar effects. Interestingly, although *Nrp1* is detected in L2/3 neurons as early as P7, innervation defects are not evident until later, perhaps because only the sum of minor changes in individual axons produces detectable phenotypes. This and other aspects of our investigation require further studies but we can propose axonal mechanisms explaining the action of Nrp1. To begin with, our findings reveal similar reductions in the growth of exuberant branches in the S1/S2 column in sh*Nrp1* and CAG-Nrp1 conditions. This allows to speculate that these phenotypes reflect opposite imbalances of synaptic stabilization/elimination at the terminal tips of developing callosal axons (*Figure 6A*). This would decrease the rate of productive axonal branching, slowing terminal arborization in S1/S2 and blocking the growth of collaterals in S2 (*Courchet et al., 2013*). In the canonical wild-type (WT) circuit, the interhemispheric axons of S1 and S2L2/3 neurons that transiently express Nrp1 succeed in establishing contralateral projections in S2. Moreover, at the population level, a higher cellular frequency of Nrp1 expression sets on the advantage of S2L2/3 neurons for innervating homotopically S2 while limiting their arborization in S1. Thus, Nrp1 dependent branching might relate to mechanisms of axonal competition. This agrees with our CTB injections in the cortical plate, which showed that injections in S1/S2 and S2 identify similar proportions of GFP$^+$ projections from S1L2/3 and S2L2/3 in IUE sh*Nrp1* and CAG-Nrp1 brains. This indicates the elimination of a competitive advantage when axons express equal levels of Nrp1.

Our results, together with previous data, support that dynamic changes in *Nrp1* expression serve multiple sequential functions. First, when callosal neurons are extending their projections (P4-P7), all SSL2/3 neurons contain low levels of *Nrp1* mRNA. Because they express Sema3A, they repel the axons of high Nrp1 expressing neurons located in the motor cortex, which serves to establish the initial dorsoventral routes of navigation (*Zhou et al., 2013*). Between P7-P10, callosal axons branch and grow collaterals in S1 in Nrp1-independent manner. During this period, SSL2/3 neurons also extend projections towards lateral domains (*Figure 6B*). Between P10 and P16, callosal axons grow exuberant branches in S1/S2 and elaborate the S2 column in manners that require the transient expression of Nrp1 – which would promote branching and the extension of these branches – (*Figure 6C*). Finally, between P16 and P30, SSL2/3 neurons downregulate Nrp1 and there is activity-dependent pruning of secondary arbors. Only those axons synapsing with coherent targets are stabilized. For some neurons, refinement ultimately results in the retraction of the main callosal projection (*Figure 6D*), as we show that sh*Nrp1* reduces the final number of S2L2/3 CPNs. Such subset of S2L2/3 neurons presumably become ipsilateral-only projecting neurons, as it occurs to WT S1L4 and half of the SSL2/3 cortical neurons in wild-types during normal postnatal refinement (*Innocenti and Clarke, 1984*; *O'Leary and Koester, 1993*; *De León Reyes et al., 2019*). Thus, altogether, within the P10-P16 window, is when the spatial and temporal differences in *Nrp1* expression weight on callosal connectivity.

Our findings highlight the spatial and temporal coordination of Nrp1 signaling required during interhemispheric wiring. Nrp1 can induce distinct signaling cascades depending on its binding to distinct ligands, such as Semaphorins and VEGF, and also on co-receptors like Plexins. It would be interesting, although out of the scope of this study, to explore the Nrp1 ligands and co-receptors involved in the late development of CC connections. the sequential role of Nrp1 in the guidance, growth, and refinement during CC circuit development likely requires the contextual use of several of such molecules. For example, Sema3A shows a high-to-low lateromedial gradient of developmental expression, triggers repulsion, and the collapse of branching points (*Kitsukawa et al., 1997*; *Zhao et al., 2011*; *Zhou et al., 2013*; *Creighton et al., 2021*). Sema3A is thus a good candidate to be involved in Nrp1 down-regulation required for terminal branching refinement in S2. On the other hand, the late postnatal functions of Nrp1 might require a developmental temporal regulation of

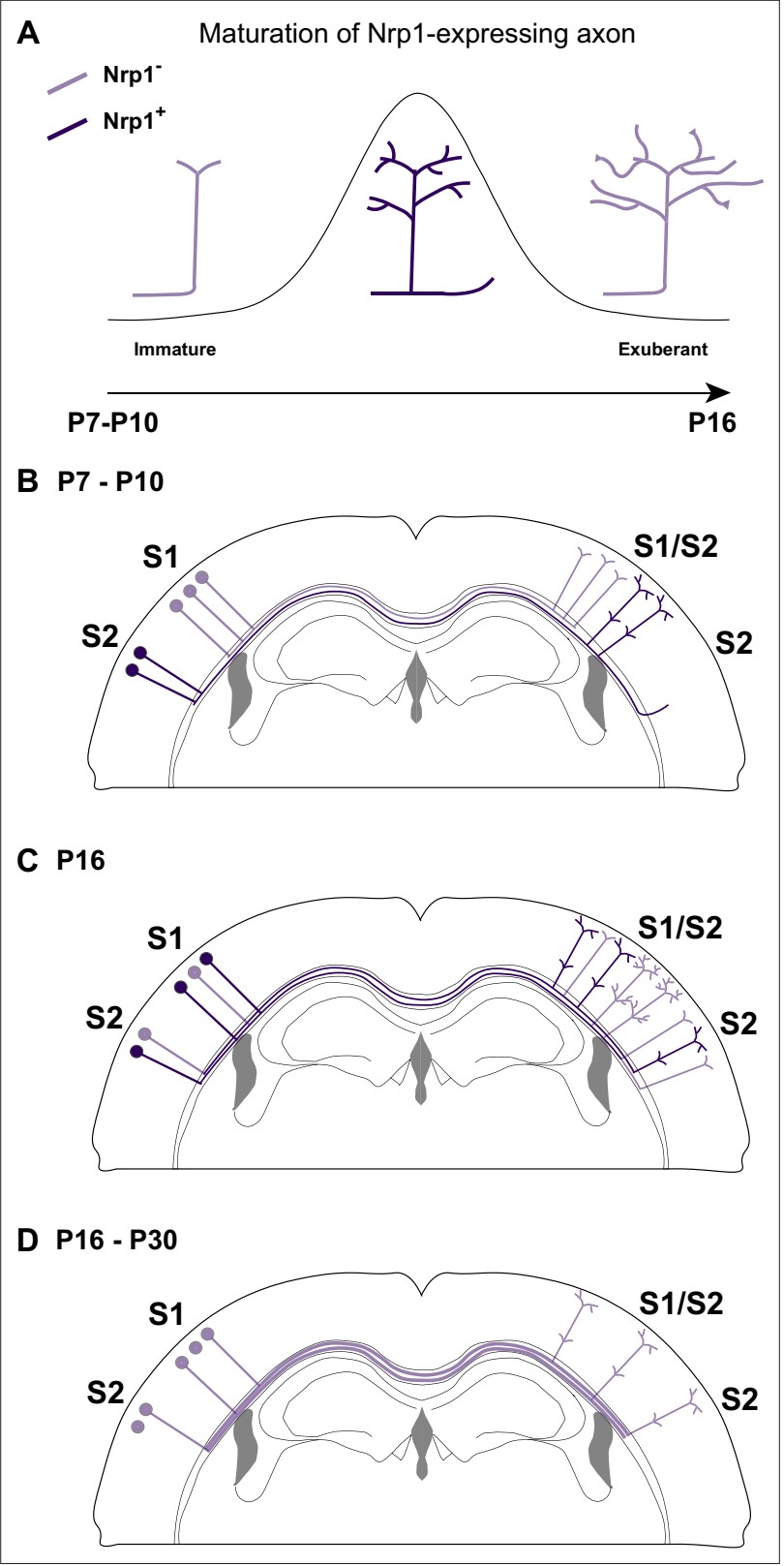

**Figure 6.** A possible model of the effects of Nrp1 transient expression in the branching and connectivity of callosal projections during development. (**A**) Transient expression of Nrp1 promotes branching at the axonal tips as well as the formation of collaterals. Both sh*Nrp1* and CAG-Nrp1 block arborization. A possible mechanistic explanation is that upregulation of Nrp1 expression stabilizes branching points and initiates the formation of secondary branches,

*Figure 6 continued on next page*

*Figure 6 continued*

while downregulation of Nrp1 allows the growth of these projections, or vice versa. (**B**) During the P7-P10 window, the S1/S2 column forms in a Nrp1-independent manner. Callosal projections from S1L2/3 present fewer branches compared to S2L2/3 neurons. CC collaterals projected by S2L2/3 axons begging to arrive at S2. (**C**) Between P10 and P16, Nrp1 expression is upregulated in S1L2/3 subpopulations and downregulated in many S2L2/3 neurons. Nrp1 upregulation promotes the growth of exuberant arbors in the S1/S2 and the S2 columns and, the formation of new collaterals. (**C**) After P16, CC axons continue their development by Nrp1-independent growth and refinement mechanisms.

its association with Plexin co-receptors. L2/3 PlexinD1 mutant neurons develop abnormal heterotopic callosal projections to the contralateral striatum, possibly due to reduced postnatal refinement (*Velona et al., 2019*). Interestingly, Nrp1 signaling for both axonal navigation and refinement in the cortex mirrors observations in the cerebellum, where Nrp1 also has a dual function. First, it guides inhibitory axons to their excitatory neuronal targets, and then, it determines the formation of synapses at specific locations within the neuronal body (*Telley et al., 2016*).

In sum, our data demonstrate that transient expression of Nrp1 regulates branching and refinement during the mid and late stages of CC development in the SS cortex. In this manner, spatial and temporal developmental differences in Nrp1 expression determine homotopic and heterotopic interhemispheric SSL2/3 connectivity between the primary and secondary areas of the somatosensory cortex.

# Materials and methods

## Key resources table

| Reagent type (species) or resource | Designation | Source or reference | Identifiers | Additional information |
|---|---|---|---|---|
| Gene (*Mus musculus.* C57BL/6 J) | Nrp1 | Genebank | Gene ID: 18,186 | |
| Strain, strain background (*Mus musculus.* Male and female) | C57BL/6JRccHsd | Envigo | | Genetic background used in all experiments |
| Transfected construct (*Aequorea victoria*) | pCAG-GFP | AddGene | Plasmid #11,150 RRID: Addgene_11150 | Plasmid construct to over-express GFP |
| Transfected construct (*Mus musculus*) | pCAG-Nrp1 | Gift from Prof. Muming Poo *Zhou et al., 2013* | | Plasmid construct to over-express Nrp1 |
| Transfected construct (*Mus musculus*) | pLKO.1 – shNrp1 | Sigma-Aldrich | ID: TRCN0000029859 | Lentiviral construct to express the sh*Nrp1*. |
| Antibody | Anti-digoxigenin-alkaline phosphatase (sheep polyclonal. IgG) | Roche | ID:11093274910 RRID: AB_514497 | 1:5,000 |
| Antibody | Anti-GFP (chicken polyclonal. IgY) | Abcam | ID: AB13970 RRID: AB_300798 | 1:500 |
| Antibody | Anti-Vglut2 (guinea pig polyclonal. Serum) | Merck | ID: AB2251 RRID: AB_2665454 | 1:500 |
| Antibody | Anti-GFP (rabbit polyclonal. IgG) | Thermofisher Scientific | ID: A11122 RRID: AB_221569 | 1:500 |
| Antibody | Anti-Chicken Alexa488 (goat polyclonal. IgY) | Thermofisher Scientific | ID: A11039 RRID: AB_142924 | 1:500 |
| Antibody | Anti-Rabbit Alexa488 (goat polyclonal. IgG) | Thermofisher Scientific | ID: A11034 RRID: AB_2576217 | 1:500 |
| Antibody | Anti-Guinea pig Alexa594 (goat polyclonal. IgG) | Thermofisher Scientific | ID: A11076 RRID: AB_141930 | 1:500 |
| Sequence-based reagent | Antisense digoxigenin-labeled | Roche | ID: 11277073910 | |

*Continued on next page*

*Continued*

| Reagent type (species) or resource | Designation | Source or reference | Identifiers | Additional information |
|---|---|---|---|---|
| Sequence-based reagent | *Nrp1* probe | Gift from V. Gil-Fernández and J.A del Río | *Mata et al., 2018* | |
| Sequence-based reagent | *Nrp1-FW* | This paper | qPCR primers | ACACAGAAATTAAAATTGATGAAACAG |
| Sequence-based reagent | *Nrp1-RV* | This paper | qPCR primers | GGATGGGATCCAGGGTCT |
| Sequence-based reagent | *GFP-FW* | This paper | qPCR primers | CAACCACTACCTGAGCACCC |
| Sequence-based reagent | *GFP-RV* | This paper | qPCR primers | GTCCATGCCGAGAGTGATCC |
| Sequence-based reagent | *Gus-FW* | This paper | qPCR primers | AGCCGCTACGGGAGTCG |
| Sequence-based reagent | *Gus-RV* | This paper | qPCR primers | GCTGCTTCTTGGGTGATGTCA |
| Peptide, recombinant protein | Subunit B of cholera toxin (CTB) conjugated to Alexa 555 | Thermofisher Scientific | ID: C34776 | Axonal retrograde labelling |
| Commercial assay or kit | NZY Total RNA isolation | NZYTech | ID: MB13402 | RNA extraction |
| Commercial assay or kit | First-strand cDNA Synthesis kit | Merck | ID: 27-9261-01 | cDNA synthesis |
| Commercial assay or kit | GoTaq qPCR Master Mix | Promega | ID: A6002 | RT-qPCR |
| Commercial assay or kit | Qiagen Plasmid Maxi Kit | Qiagen | ID: 12,165 | Plasmid DNA purification Kit |
| Chemical compound, drug | PFA (paraformaldehyde) | Merck | ID:1.04005.1000 | Tissue fixation |
| Chemical compound, drug | PBS 10 X (phosphate buffer saline) | iNtRON Biotechnology | ID: 102,309 | |
| Chemical compound, drug | Sucrose | Merck | ID:1.07651.1000 | Tissue cryoprotection |
| Chemical compound, drug | Sucrose | Merck | ID: S0389 | Tissue cryoprotection |
| Chemical compound, drug | Formalin solution, neutral buffered, 10% | Sigma-aldrich | ID: HT501128-4L | Tissue fixation |
| Chemical compound, drug | Deionized formamide | Millipore | ID: S4117 | |
| Chemical compound, drug | Denhardst's 1 X | Sigma-Aldrich | ID: D2532 | |
| Chemical compound, drug | Dextran sulphate 10 X | Sigma-Aldrich | ID: 4,911 | |
| Chemical compound, drug | tRNA | Sigma-Aldrich | ID: R6625 | |
| Chemical compound, drug | Blocking solution | Roche | ID: 11096176001 | |
| Chemical compound, drug | Hoechst 33,342 | Invitrogen | ID: H1399 | Nuclei staining |
| Chemical compound, drug | 4',6-diamidino-2-phenylindole (DAPI) | Merck | ID: D9542 | Nuclei staining |
| Chemical compound, drug | PBS·DEPC | Sigma-aldrich | ID: D5758 | |
| Chemical compound, drug | O.C.T Tissue-Tek compound | Sakura Tissue-Tek | ID: 4,583 | Freeze solution |
| Software, algorithm | Graphpad Prism 8 | Graphpad Software | RRID:SCR_002798 | Statistical software |
| Software, algorithm | Fiji-ImageJ | Fiji | Schindelin, J. et al. 2012. RRID:SCR_003070 | Imaging software |
| Software, algorithm | Semi-automated counting cells macros | This paper | GitHub: https://github.com/FMartin30/Macros; *Bragg-Gonzalo, 2022* | Macros to semi-automated counted of GFP+ cells |
| Other | Fetal Bovine Serum (FBS) | Thermofisher Scientific | ID: A31605 | Blocking solution for immunofluorescence |

## Animals

Wild-type (WT) C57BL/6JRccHsd (Envigo Laboratories, formerly Harlan. Indianapolis, the U.S.) mice were used in all experiments. The morning of the day of the appearance of a vaginal plug was defined as embryonic day 0.5 (E 0.5). Animals were housed and maintained following the guidelines from the European Union Council Directive (86/609/European Economic Community). All procedures for handling and sacrificing complied with all relevant ethical regulations for animal testing and research.

All experiments were performed under the European Commission guidelines (2010/63/EU) and were approved by the CSIC and the Community of Madrid Ethics Committees on Animal Experimentation in compliance with national and European legislation (PROEX 124–17; 123–17).

## In situ hybridization

P4 and P7 brains were fixed in 4% paraformaldehyde (PFA) (#1.04005.1000, Merck. Darmstadt. Germany) diluted in phosphate buffer (PBS 1 X) for 2 and 4 hr at room temperature (RT), respectively. P16 and P56 brains were collected from animals perfused intracardially with sterile PBS 1 X followed by 4% PFA and post-fixed with 4% PFA overnight (O/N) at 4 °C. After fixation, brains were PBS 1 X washed and cryoprotected in 15% sucrose (#1.07651.1000, Merck. Darmstadt. Germany)/PBS 1 X. Lastly, they were embedded in 7.5% gelatin (#G2625, Sigma. Merck. Darmstadt. Germany)/15% sucrose (#1.07651.1000, Merck. Darmstadt. Germany)/PBS1X and frozen at –80 °C. Coronal cryostat sections were cut at 16 μm thickness.

In situ hybridization (ISH) was carried out as previously described by *Di Meglio et al., 2013*. Briefly, the tissue was incubated with 4% PFA for 10 min at 4 °C. Then, prehybridization was performed at RT with hybridization buffer (50% deionized formamide (#S4117, Millipore, Merck), SALTS 1 X, Denhardt's 1 X (#D2532, Sigma, Merck. Darmstadt. Germany), 10% dextran sulphate (#4911, Sigma, Merck), tRNA 1 mg/ml (#R6625, Sigma, Merck)) for 1 hr in a humified chamber with 5 x SSC and 50% deionized formamide (#S4117, Millipore, Merck). Tissue sections were incubated with the anti-sense digoxigenin-labeled (#11277073910, Roche, Merck. Darmstadt. Germany) probe (0.25 ng/μl in hybridization buffer) O/N at 72 °C. Following hybridization, the slides were washed in SSC 0.2 X for 90 minutes at 72 °C and then blocked in 2% blocking solution (#11096176001, Roche, Merck. Darmstadt. Germany) in MABT at pH 7.5 (maleic acid 500 mM, NaOH 1 M, NaCl 750 mM, 0.1%Tween-20) for 1 hr at RT and then incubated O/N at 4 °C with anti-digoxigenin-alkaline phosphatase antibody (#11093274910, Roche, Merck. Darmstadt. Germany) at a 1:5000 diluted in MABT. After several washes, the alkaline phosphatase activity was developed using NBT and BCIP diluted in NTMT solution at pH 9.5 (Tris 100 mM, NaCl 100 mM, MgCl2 50 mM, 0.1% Tween-20) for 20 hr at RT. The *Nrp1* probe was kindly provided by V. Gil-Fernández (*Mata et al., 2018*).

For double ISH and immunofluorescence (IF) staining, ISH was carried out prior to IF as previously described (*Di Meglio et al., 2013*). The following primary antibodies were used: chicken anti-GFP (#AB13970, Abcam. Cambridge. UK) and guinea pig anti-VGlut2 (#AB2251, Merck. Darmstadt. Germany) followed by the secondary antibodies: goat anti-chicken-Alexa 488 (#A11039, Life Technologies, Thermo Fisher Scientific. Massachusetts, the U.S.) and goat anti-guinea pig-Alexa 594 (#A11076, Life Technologies, Thermo Fisher Scientific. Massachusetts, the U.S.), respectively. Hoechst 33,342 (#H1399, Invitrogen, Thermo Fisher Scientific. Massachusetts, the U.S.) was used for nuclei counter-staining.

## In utero electroporation and plasmids

Plasmids used were pCAG-GFP (Addgene, plasmid #11150), pCAG-Nrp1 (gift from Prof. Mu-ming Poo), and sh*Nrp1* in pLKO.1 vector (hairpin sequence: CCTGCTTTCTTCTCTTGGTTTC. #TRCN0000029859, Merck. Darmstadt. Germany). In utero electroporation was performed as previously described (*Briz et al., 2017*). Briefly, a mixture of the specified plasmids at a concentration of 1 μg/μl each (pCAG-GFP or pCAG-Nrp1) or 0.6 μg/μl (pLKO.1-sh*Nrp1*) was injected into the embryo's left lateral ventricle using pulled glass micropipettes. Five voltage pulses (38 mv, 50ms) were applied using external paddles oriented to target the somatosensory cortex or anterior cingulate cortex. After birth, P2 GFP[+] pups were selected and allowed to develop normally until P10, P14, and P28. After sectioning, analyses were performed only in animals in which the electroporated area included both S1 and S2.

## Quantitative RT-PCR analysis

The tissue was freshly collected from electroporated mouse brains and homogenized with 3 mm stainless steel Lysing beads (Alpha Nanotech, VWR. Pennsylvania, the U.S.) in PBS·DEPC (diethyl pyrocarbonate. #D5758. Sigma-Aldrich. Merck. Darmstadt. Germany) for 1 min at 30 Hz with a TissueLyser (MM300, Retsch. Düsseldorf. Germany). Total RNA was isolated using NZY Total RNA Isolation kit (#MB13402, Nzytech. Lisbon. Portugal) following the manufacturer's instructions. cDNA was obtained

from 1 µg of total RNA with First-Strand cDNA Synthesis kit (#27-9261-01, GE, Merck. Darmstadt. Germany) in a 15 µl reaction volume.

Quantitative real-time qRT-PCR was performed using GoTaq qPCR Master Mix (#A6002, Promega. Wisconsin, the U.S.) following the protocol of the manufacturer in a CFX-384 Touch Real-Time PCR Detection System (BioRad. California, the U.S.). The following gene-specific primer pairs were used: *Nrp1*-Forward 5'-ACACAGAAATTAAAATTGATGAAACAG-3', *Nrp1*-Reverse 5'-GGATGGGATCCAGGGTCT-3', *GFP*-Forward 5'-CAACCACTACCTGAGCACCC-3', *GFP*-Reverse 5'-GTCCATGCCGAGAGTGATCC-3', *Gus*-Forward 5'-AGCCGCTACGGGAGTCG-3' and *Gus*-Reverse 5'-GCTGCTTCTTGGGTGATGTCA-3'.

*Nrp1* and *GFP* expression levels were quantified in triplicates and normalized to *Gus* expression levels. Resultant data were analyzed using the comparative Ct method.

## CTB injections for retrograde labeling

Retrograde labeling from the CC and the cortical plate was performed by injecting subunit B of cholera toxin (CTB) conjugated to Alexa Fluor 555 (#C-34776, ThermoFisher Scientific. Massachusetts, the U.S.). Injections were performed in the CC, close to the midline, as previously reported (*De León Reyes et al., 2019*), or in the cortical plate; in both cases, in the contralateral non-electroporated hemisphere (right hemisphere). Stereotaxic coordinates, injection volumes, and procedures for different developmental stages for injections in CC were performed as previously described (*De León Reyes et al., 2019*). For cortical plate injections at P30, stereotaxic coordinates (anteroposterior (AP), mediolateral (ML), and dorsoventral (DV) axes from Bregma) were adjusted using the atlas of Paxinos (*Paxinos and Franklin, 2004*) and used as follow: S1/S2 injections (–1.34 mm AP; + 3.7 mm ML; –0.4 ~ –0.5 mm DV) and, S2 injections (–1.34 mm AP; + 3,7 mm ML; –0.7 ~ –0.8 mm DV); injecting 100 nL of CTB solution at 4 nl s$^{-1}$. Animals were anesthetized during the surgical procedure with isoflurane/oxygen and placed on a stereotaxic apparatus (Harvard Apparatus. Massachusetts, the U.S.). CTB particles (diluted at 0.5% in phosphate-buffered saline (PBS)) were injected with a Drummond Nanoject II Auto-Nanoliter Injector using 30 mm pulled glass micropipettes (3000205 A and 3000203 G/X. Drummond Scientific Co. Pennsylvania, the U.S.). Mice were intrapericardially perfused with formalin two days after the surgery and brains were extracted and fixed O/N in formalin at 4 °C. After fixation, brains were cryoprotected with 30% sucrose (#S0389. Merck. Darmstadt. Germany) and frozen in Tissue-Tek O.C.T. Compound (#4583, Sakura Tissue-Tek. Tokyo. Japan).

## Immunohistochemistry

Fifty µm free-floating brain cryosections were used for immunofluorescence. Rabbit polyclonal anti-GFP (#A11122, Thermo Fisher Scientific. Invitrogen. Massachusetts, the U.S.) was used as primary antibody and goat anti-rabbit-Alexa 488 (#A11034, Thermo Fisher Scientific. Life Technologies. Massachusetts, the U.S.) as the secondary antibody. Nuclei were stained with 4',6-diamidino-2-phenylindole (DAPI) (#D9542, Merck. Darmstadt. Germany).

## Imaging

In situ hybridization chromogenic and immunofluorescence images were obtained with a DMCTR5000 microscope equipped with a DFC500 color camera (Leica. Wetzlar. Germany). Confocal microscopy was performed using a TCS-SP5 (Leica. Wetzlar. Germany) Laser Scanning System on Leica DMI8 microscopes. Up to 50 µm optical z-sections were obtained by taking 3.5 µm serial sections with LAS AF v1.8 software (Leica. Wetzlar. Germany). Tilescan mosaic images were reconstructed with Leica LAS AF software. All images were acquired using a 512 × 512 scan format with a 20 x objective. All coronal sections correspond to rostro-caudal coordinates between Bregma −0.82,−1.46 mm (*Paxinos and Franklin, 2004*). For the acquisition and quantifications of the fluorescence signal (*Rodríguez-Tornos et al., 2016*; *Briz et al., 2017*), detectors were set to ensure equivalent threshold and signal-to-noise ratios between all samples. The maximum threshold signal was set by ensuring that no pixels were saturated. The threshold for background noise was determined using regions outside of the electroporated area (*Rodríguez-Tornos et al., 2016*; *Briz et al., 2017*). This approach ensures linearity between samples.

## Counting of ipsilateral electroporated neurons and fluorescence quantification

Quantification of the total number of electroporated neurons was done in a semi-automated manner using an ImageJ custom macro written in Java (https://github.com/FMartin30/Macros, copy archived at swh:1:rev:290118c15f4bd80e241fd3090035432afc5e0edb; *Bragg-Gonzalo, 2022*). First, the total electroporated region in S1 and S2 was outlined and cortical layers separated based on their distinct cell densities. L1 being a sparsely populated layer while the border between layers 2/3 and 4 was determined by the higher nuclei density of L4. The threshold was set to identify neuronal somas and the cell numbers in each layer was obtained using the script. The selected regions of interest were then manually checked. All analyses were conducted in blind conditions.

Quantification of innervation was performed in tilescan images of electroporated (ipsilateral) and non-electroporated (contralateral) hemispheres. The values in selected areas were measured manually delimiting ROIs, adjusting the threshold above the noise (making a binary image), and measuring the integrated density (using Fiji-ImageJ *Schindelin et al., 2012*). The measures of contralateral ROIs were normalized to the value in the ipsilateral area of the same coronal section to avoid differences in electroporation efficiency. To confirm the results, contralateral normalizations without considering the ipsilateral signal were calculated as an alternative method. To quantify CC fasciculation, we measure the fluorescence profile throughout ten equal distance bins of ROI at the midline. The different profiles were plotted to identify changes in dorsoventral routes.

## Callosal neurons (CTB$^+$) quantification

Quantification of CTB$^+$ over GFP$^+$ cells in the primary (S1) and secondary (S2) somatosensory areas was performed on single plane confocal images from z-stacks (*De León Reyes et al., 2019*). The proportions of CTB$^+$ cells were calculated among randomly selected GFP$^+$ cells in the ipsilateral (electroporated) hemisphere. For quantification of GFP$^-$ populations, the proportions of CTB$^+$ cells were calculated over randomly selected DAPI$^+$ cells, excluding GFP$^+$ cells. Functional areas of the adult mouse brain were identified using the atlas of Paxinos (*Paxinos and Franklin, 2004*).

## Statistical analysis

The sample size was determined to be adequate based on the magnitude and consistency of measurable differences between groups. Each experimental condition was carried out with a minimum of three biological replicates, a minimum of two sections from each brain, and included a minimum total number of 300 counted cells. During experiments, investigators were not blinded to the electroporation condition of animals. Results are expressed as the mean ± standard error of the mean (SEM). Results were compared using two-way ANOVA and one-way ANOVA with Tukey Posthoc comparison. Statistical tests were performed using Prism eight software (GraphPad Software. California, the U.S.). The source data underlying main *Figures 2–5*, and figures supplement, are provided as a Source Data files.

## Acknowledgements

We are grateful to R Gutierrez, A Morales, S Gutiérrez-Erlandsson, and A Oña for technical assistance. J García-Marqués, LA Weiss, N S de León, I Varela, and E Marcos for critical reading and advice. F Martín-Fernández holds an FPU fellowship from the Spanish MEFP, FPU15/02111. A Bermejo-Santos holds an FPI fellowship from the Spanish MCIN, PRE19-089366. L Bragg-Gonzalo holds a fellowship from the European Union Horizon 2020 research and innovation program under the Marie Sklodowska-Curie grant agreement No. 713,673 and "La Caixa" Foundation (ID 100010434, the fellowship code is LCF/BQ/IN17/11620044). C García-Briz was supported by a fellowship from the Spanish MICINN, FPI-BES-2012–056011 by MCIN/AEI/ 10.13039/501100011033 and by "ESF Investing in your future". E Serrano-Sainz holds a Ramón y Cajal Contract (RyC-2016-20537). This work was funded by PID2020-112831GB-I00 MCIN/ AEI /10.13039/501100011033/, and by SAF2017-83117-R and RED2018-102553T funded by MCIN/ AEI /10.13039/501100011033/ and "*FEDER Una manera de hacer Europa*" by the European Union.

# Additional information

## Funding

| Funder | Grant reference number | Author |
| --- | --- | --- |
| Ministerio de Educación y Formación Profesional | FPU15/02111 | Fernando Martín-Fernández |
| Ministerio de Ciencia e Innovación | FPI PRE19-089366 | Ana Bermejo-Santos |
| H2020 Marie Skłodowska-Curie Actions | 713673 | Lorena Bragg-Gonzalo |
| "la Caixa" Foundation | 100010434 | Lorena Bragg-Gonzalo |
| Ministerio de Ciencia e Innovación | FPI-BES-2012-05601 | Carlos G Briz |
| Ministerio de Ciencia e Innovación | PID2020-112831GB-I00 | Marta Nieto Lopez |
| Ministerio de Ciencia e Innovación | RED2018-102553T | Marta Nieto Lopez |
| Ministerio de Ciencia e Innovación | SAF2017-83117-R | Marta Nieto Lopez |
| Ministerio de Ciencia e Innovación | RyC-2016-20537 | Esther Serrano-Saiz |

The funders had no role in study design, data collection and interpretation, or the decision to submit the work for publication.

## Author contributions

Fernando Martín-Fernández, Conceptualization, Data curation, Formal analysis, Investigation, Methodology, Writing – review and editing; Ana Bermejo-Santos, Formal analysis, Investigation, Methodology, Validation, Visualization; Lorena Bragg-Gonzalo, Formal analysis, Methodology, Validation, Writing – review and editing; Carlos G Briz, Conceptualization, Supervision; Esther Serrano-Saiz, Conceptualization, Methodology, Writing – review and editing; Marta Nieto, Conceptualization, Formal analysis, Funding acquisition, Investigation, Methodology, Project administration, Supervision, Writing – original draft, Writing – review and editing

## Author ORCIDs

Fernando Martín-Fernández  http://orcid.org/0000-0003-4060-0118
Ana Bermejo-Santos  http://orcid.org/0000-0002-2595-0729
Lorena Bragg-Gonzalo  http://orcid.org/0000-0001-6848-4556
Esther Serrano-Saiz  http://orcid.org/0000-0003-0077-878X
Marta Nieto  http://orcid.org/0000-0002-8349-8435

## Ethics

Animals were housed and maintained following the guidelines from the European Union Council Directive European Economic Community. All procedures for handling and sacrificing complied with all relevant ethical regulations for animal testing and research. All experiments were performed under the European Commission guidelines (2010/63/EU) and were approved by the CSIC and the Community of Madrid Ethics Committees on Animal Experimentation in compliance with national and European legislation (PROEX 124-17; 123-17).

## Decision letter and Author response

Decision letter https://doi.org/10.7554/eLife.69776.sa1
Author response https://doi.org/10.7554/eLife.69776.sa2

## Additional files

### Supplementary files
• Transparent reporting form

### Data availability
All data generated or analysed during this study are included in the manuscript and supporting files. Source data files have been provided for all figures.

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
