## [Editor Report]

Your study highlights a novel role of Neuropilin 1 in regulating callosal connectivity at the level of the areal map with important insights on how areas mature and develop. The revisions of your manuscript have now clarified some of the methodological issue and we believe that it will be an important contribution to the field.

---

## [Decision Letter]

**Decision letter after peer review:**

Thank you for submitting your article "Role of Nrp1 in controlling cortical interhemispheric circuits" for consideration by *eLife*. Your article has been reviewed by 3 peer reviewers, and the evaluation has been overseen by a Reviewing Editor and John Huguenard as the Senior Editor. The reviewers have opted to remain anonymous.

The Reviewing Editor has drafted this to help you prepare a revised submission. Your study highlights a novel role of Neuropilin 1 in regulating callosal connectivity at the level of the areal map with important insights on how areas mature and develop. However, several reviewers have stressed the need for additional controls and experiments, in particular to address the following points:

Essential revisions:

– All reviewers underlined the importance to characterize in more depth the effects of Nrp1 ShRNA electroporation, in terms of impact on progenitors/ migration and on the level/timing of Nrp1 expression. It would thus be essential to address this point in the revised manuscript.

– Several reviewers stressed the need to clarify and change some of the quantifications and normalisations that should be taken into account (number of sections analyzed versus number of mice, quantifications of the GFP-positive cells, inter-individual variability).

– Since *eLife* targets a broad audience, it would be important to modify the manuscript to make it more accessible for non-specialists and clarify some statements, as mentioned by two of the reviewers.

We hope that these comments help you in your revisions and look forward to receiving your manuscript.

*Reviewer #1 (Recommendations for the authors):*

I have the following questions and suggestions to the authors that would solidify and/or clarify the conclusions made from the study:

1) There is previous evidence that Nrp1/Sema3A signalling may be involved in the migration of cortical neurons, with the loss of Nrp1 perhaps affecting this process to a greater extent than overexpression (Chen et al., 2007; Hatanaka et al., 2009). This, combined with evidence that shRNAs can sometimes cause off-target migration defects (Baek et al., 2014) makes it possible that some of the effects observed in the manipulations of this paper may be due to migration defects. I suggest cell counts of the electroporated hemisphere with a comparison of the distributions by bins/cortical layer at P16 and P30 to better understand whether changes in migration might underlie some of the branching and axon guidance phenotypes.

2) In the final Results section, the authors present an interpretation: "Refinement of L2/3 CPNs, in both S1 and S2 areas, was not altered upon overexpression of Nrp1. However, electroporation of shNrp1 modified refinement (Figure 5 D-I). In P16 control brains, 50% of GFP+ S1L2/3 neurons were CTB+ (CPNs) (Figure 5D and E). This number increased to 65% in shNrp1-targeted S1L2/3 (Figure 5D and F), indicating that low Nrp1 expression delays axonal refinement." An alternative interpretation of this is that the Nrp1 gradient determines to some extent the number and location of neurons initially projecting across the midline. This could be feasible, given that lowering the expression of Nrp1 in S1 neurons may make them more "lateral-S1-like" in identity, and in wildtype brains the lateral S1 neurons end up more heavily projecting through the corpus callosum. I understand it may be difficult to show this experimentally, as it would necessitate earlier ages of collection that have not been performed, but at least an allusion to such an alternative interpretation would be appreciated.

3) In figure 4, it is unclear how the shaded areas of significant difference in panels E, I, J, K and L have been determined. The posthoc with Tukey's test p values reported in the legend and signified by the asterisks and hashes are presumably following the two-way ANOVA and comparing the main effect of experimental condition (across all bins), so how were the shaded areas determined?

4) Figure 4 in general shows fairly subtle changes in midline distribution of the electroporated axon tract, which could be easily found by very slight alterations in the positioning of the electroporated cell field, especially given the n=3 animal numbers of this experiment. I therefore suggest a quantification of electroporated cell field's mediolateral extent and position, as a comparable size and positioning between conditions is particularly relevant to this figure, but also for figures 1 and 3.

5) At times the phrasing of results and conclusions could be clarified as it is not always apparent what is meant, for instance "They cannot be merely explained by selective repulsion from the cortical plate, although they do not discard its contribution" (line 268) and "This again that not support axonal repulsion, may suggest a disadvantage at a possible axonal competition" (line 305). I suggest clarifying the language throughout the manuscript, as well as potentially adding more information to the summary figure 6 such that it encapsulates the other manuscript findings (midline refinement, ipsilateral cell populations etc). There is a lot of very complex data in here that is potentially valuable to the field, but at the moment it may be difficult for a non-expert reader to access it in totality and thereby appreciate the novel advances.

*Reviewer #2 (Recommendations for the authors):*

While this is a potentially interesting study, the results are still very preliminary and superficial. Moreover, serious technical considerations limit interpretation of the results at this stage, such that the value of the message conveyed is unclear.

Technical concerns: There are multiple severe technical concerns to this study which in our view preclude publication if not directly and fully addressed.

1. The quantifications were not blindly performed (see Methods). Given the high experimental variability of the techniques used in this study (in utero electroporations and stereotaxic injections), analyses should be performed in a strictly blind setting.

2. Normalization of the GFP signal in the ROIs should be done using the number of electroporated cells rather than the GFP signal in the electroporated site, which is highly dependent on the acquisition settings.

3. On all graphs, the dots correspond to the sections analyzed and not the brains, so that the interindividual variability cannot be assessed. For all graphs and statistical analyses, the average of all sections for each brain should be used and not the value of each section. This is a critical point as it misleads the reader into assuming high power of analysis. At the very least three animals per condition (ideally at least 5) should be used to provide convincing conclusions. In addition, given the variability of GFP signal upon in utero electroporation, taking only 2 sections per brain seems very low. At least 3 to 4 sections should be quantified and averaged to avoid any bias.

4. For CTB injections, the injection sites for all brains should be shown (or a quantification of VPM / Po cell body location) in the supplementary figure.

5. For the CC analysis at the midline, is the dorso-ventral distribution similar all along the rostro-caudal axis? How is this parameter taken into account in the analyses? Are all electroporated sites equivalent in term of rostro-caudal position? This should be quantified, and the data should be normalized for this parameter.

Other concerns

1. Timing of down / up regulation of Nrp1 expression

Here the shRNA against Nrp1 was electroporated at E15.5: is the down-regulation still efficient when callosal axons are extending and invading the contralateral cortex (ie. in the first postnatal weeks)? The levels of Nrp1 should be addressed in electroporated neurons, for example by RT-qPCR or western blot after FACS (this would allow quantifying differences of expression between S1 and S2 – see discussion l.290-291).

The neuropilin-semaphorin family was shown to play a role at the progenitor levels (Castellani lab). Since the constructs used here induce constitutive gain or loss of function, how can the authors be sure that the previous steps of development are not affected in L2/3 neurons?

2. Role of Nrp1 in branching

This is not directly addressed in the study and should be removed from the text. To formally conclude on this aspect, the authors should use sparse electroporation and morphological reconstruction of contralateral axons.

3. Biological mechanism underlying the findings

Neuropilins do not act alone and need co-receptors of the plexin family. While this is an important point to understand the mechanisms, this is only superficially mentioned in the discussion (l. 317).

Sema3A is proposed as a likely candidate for Nrp1 late function in branching. This could be addressed using Sema3A gain or loss of function.

What could control Nrp1 expression levels at these postnatal stages? This should be mentioned in the discussion.

*Reviewer #3 (Recommendations for the authors):*

The paper is a difficult read for non-specialists in the field. I suggest simplifying the language for the sake of broader audience. The final diagram might include postnatal refinement data.

---

## [Author Response]

Essential revisions:– All reviewers underlined the importance to characterize in more depth the effects of Nrp1 ShRNA electroporation, in terms of impact on progenitors/ migration and on the level/timing of Nrp1 expression. It would thus be essential to address this point in the revised manuscript.

We have characterized in-depth the expression of Nrp1 through postnatal development to understand the effects of our manipulations. We have confirmed that shNrp1 or CAG-Nrp1 do not produce migration defects. We have also confirmed that the effects of shNrp1 in P16 brains are detectable after electroporation at E15.5 and that our manipulations of Nrp1 do not alter migration or laminar position. Importantly, the characterization of postnatal expression revealed unreported temporal and spatial patterns. In light of this, we can now offer a clearer interpretation of our data. We consider these additions to be an important improvement of the manuscript and thank the reviewers for these important comments.

– Several reviewers stressed the need to clarify and change some of the quantifications and normalisations that should be taken into account (number of sections analyzed versus number of mice, quantifications of the GFP-positive cells, inter-individual variability).

We have made newly blinded quantifications and changed the normalization method to the number of electroporated cells. None of these change the results, thus validating our previous methods. In fact, the quantifications relative to the number of cells suggested by reviewer #2 increase precision. This increases the statistical power of the measurements. All the data remains the same except that the statistical analysis with the new quantification produces lower p-values in the comparisons of the measurements of the effects of shNrp1 in the S2 column. The data normalized to the GFP fluorescence signal in the ipsilateral electroporated hemisphere from the previous version is now shown as a supplement. Previously, mean data showed the same tendency towards reductions, but the statistical analyses showed *p* values that did not allow discarding that the data was equal or different from controls. We also responded to the concerns about inter-individual variability, rechecked that this is not the source of our phenotypes, and corrected that each value of the analysis represents the average data from individual mice and not the two measurements of each section.

– Since eLife targets a broad audience, it would be important to modify the manuscript to make it more accessible for non-specialists and clarify some statements, as mentioned by two of the reviewers.

We have rewritten major parts of the manuscript and changed Figure 6 summarizing the data. We hope we have improved this aspect.

Reviewer #1 (Recommendations for the authors):I have the following questions and suggestions to the authors that would solidify and/or clarify the conclusions made from the study:1) There is previous evidence that Nrp1/Sema3A signalling may be involved in the migration of cortical neurons, with the loss of Nrp1 perhaps affecting this process to a greater extent than overexpression (Chen et al., 2007; Hatanaka et al., 2009). This, combined with evidence that shRNAs can sometimes cause off-target migration defects (Baek et al., 2014) makes it possible that some of the effects observed in the manipulations of this paper may be due to migration defects. I suggest cell counts of the electroporated hemisphere with a comparison of the distributions by bins/cortical layer at P16 and P30 to better understand whether changes in migration might underlie some of the branching and axon guidance phenotypes.

We thank the reviewer for this constructive criticism. We have performed these analyses and counted the GFP positive cells throughout the radial axis of the cortex and plotted cell distributions in bins as suggested. The data show that our knocking down constructs do not alter the normal migration of neurons. The data is shown in Figure 2 figure supplement 3 and Figure 4 figure supplement 1. Chen et al. (2008) reported experiments showing that electroporation of RNAi Nrp1 constructs in rats and mice block radial migration. They also showed that the conditional deletion of exon 2 of Nrp1 in mice by in utero electroporation produced the same phenotypes. However, using the same mouse Nrp1 conditional line, the studies by Zhou et al. 2013, report no defects in migration. Similarly, Hatanaka et al., 2009 found that overexpression of a dominant-negative form of Nrp1 disrupts axonal development but not the migration of cortical excitatory neurons. We do not observe defects in migration and cannot explain the discrepancies or our and others with the studies original findings in Chen et al., 2008.

2) In the final Results section, the authors present an interpretation: "Refinement of L2/3 CPNs, in both S1 and S2 areas, was not altered upon overexpression of Nrp1. However, electroporation of shNrp1 modified refinement (Figure 5 D-I). In P16 control brains, 50% of GFP+ S1L2/3 neurons were CTB+ (CPNs) (Figure 5D and E). This number increased to 65% in shNrp1-targeted S1L2/3 (Figure 5D and F), indicating that low Nrp1 expression delays axonal refinement." An alternative interpretation of this is that the Nrp1 gradient determines to some extent the number and location of neurons initially projecting across the midline. This could be feasible, given that lowering the expression of Nrp1 in S1 neurons may make them more "lateral-S1-like" in identity, and in wildtype brains the lateral S1 neurons end up more heavily projecting through the corpus callosum. I understand it may be difficult to show this experimentally, as it would necessitate earlier ages of collection that have not been performed, but at least an allusion to such an alternative interpretation would be appreciated.

Thank you for this suggestion and this comment. In reading this comment we realized that we were not sufficiently clear on explaining how our previous publication (De Leon-Reyes et al., 2019) changes the knowledge of how callosal fates are acquired. Our publication demonstrated that callosal fate is discarded rather than obtained only by some populations. Virtually, all L2/3 neurons are CPN at birth and we demonstrated that they all project across the midline. Hence, Nrp1 expression in WT cannot affect the number and location of neurons initially projecting across the midline. We have explained this more concisely and boldly in the introduction- pg 4, line 67-74 and result section-pg 10-11 ln 271-279. We apologize for the confusion.

3) In figure 4, it is unclear how the shaded areas of significant difference in panels E, I, J, K and L have been determined. The posthoc with Tukey's test p values reported in the legend and signified by the asterisks and hashes are presumably following the two-way ANOVA and comparing the main effect of experimental condition (across all bins), so how were the shaded areas determined?

The shaded areas represented segments where Tukey´s test values showed significance for continuous adjacent bins. Nevertheless, because the reviewer was right in pointing out that we should revise if the lateral to the medial position of the cells was a factor accounting for the observed differences, we have modified this figure and moved this figure to a supplementary figure- (see below).

4) Figure 4 in general shows fairly subtle changes in midline distribution of the electroporated axon tract, which could be easily found by very slight alterations in the positioning of the electroporated cell field, especially given the n=3 animal numbers of this experiment. I therefore suggest a quantification of electroporated cell field's mediolateral extent and position, as a comparable size and positioning between conditions is particularly relevant to this figure, but also for figures 1 and 3.

We are very thankful to the reviewer for this observation. We have performed quantifications of the distributions of the electroporated cells in S1 and S2. These confirmed that we analyze brains with similar distributions in S1 and S2 of electroporated brains (Figure 2 supplement 3 and Figure 4 supplement 1). We statistical tests demonstrated that distributions in S1 and S2 are not a source of variability when concerning innervation of the cortical plate. Differences in innervation are accounted for only by the variable condition (control, CAG-Nrp1, and shNrp1) (p values of the different interactions shown in the figure legend). This is in agreement with all neurons behaving in an equal way when they express similar levels of Nrp1 when electroporating shNrp1 and CAG-Nrp1 constructs. However, minor changes in the exact number of cells in medial and lateral positions are unavoidable and the reviewer was right, these analyses showed that minor differences in the distributions of electroporated neurons explain the subtle changes in the distribution of axons at the midline we reported in previous figure 4. We, therefore, performed a new analysis with only the brains with the most similar distributions along the lateral to the medial axis (new Figure 4 figure supplement 3). This analysis shows no differences between conditions. This is in agreement with that the location of the projecting neuron in the somatosensory has a greater impact on the dorso-ventral level path of callosal projection than its level of Nrp1 expression, possibly because it is determined by Sema3A expression -which repels the Nrp1 expressing axons of motor neurons. It also makes sense because now we show that in SS L2/3 neurons – Nrp1 expression is not activated until the post-crossing stages.

The analysis at the midline is valid and it still serves to demonstrate that shNrp1 and CAG-Nrp1 projections cross the midline using the ventral path as control axons, as was also concluded in the previous version but does not support refinement. We sincerely thank the reviewer for correcting us on this.

5) At times the phrasing of results and conclusions could be clarified as it is not always apparent what is meant, for instance "They cannot be merely explained by selective repulsion from the cortical plate, although they do not discard its contribution" (line 268) and "This again that not support axonal repulsion, may suggest a disadvantage at a possible axonal competition" (line 305). I suggest clarifying the language throughout the manuscript, as well as potentially adding more information to the summary figure 6 such that it encapsulates the other manuscript findings (midline refinement, ipsilateral cell populations etc). There is a lot of very complex data in here that is potentially valuable to the field, but at the moment it may be difficult for a non-expert reader to access it in totality and thereby appreciate the novel advances.

We appreciate this comment. We have edited the manuscript trying to avoid long complex wording. We have tried to summarize the information of the model in the discussion and also changed the figure of the model (Figure 6), also because certain interpretations have been revised.

Reviewer #2 (Recommendations for the authors):While this is a potentially interesting study, the results are still very preliminary and superficial. Moreover, serious technical considerations limit interpretation of the results at this stage, such that the value of the message conveyed is unclear.

We would like to thank the reviewer for his/her suggestions. We have performed control analysis and changed quantifications. The overall results do not change. The added description of Nrp1 expression across postnatal stages sharpen the interpretation of the results. We think this improves the manuscript.

Technical concerns: There are multiple severe technical concerns to this study which in our view preclude publication if not directly and fully addressed.1. The quantifications were not blindly performed (see Methods). Given the high experimental variability of the techniques used in this study (in utero electroporations and stereotaxic injections), analyses should be performed in a strictly blind setting.

We have repeated all quantifications in blind conditions and automated some measurements (performed by new authors) and the results do not change.

2. Normalization of the GFP signal in the ROIs should be done using the number of electroporated cells rather than the GFP signal in the electroporated site, which is highly dependent on the acquisition settings.

We previously described and validated our method of quantifications using GFP signal in the ipsilateral hemisphere and describe how to set conditions that ensure that the GFP signal is proportional to the number of cells (Rodriguez-Tornos et al., 2016 and Briz et al., 2017). This is the method we had employed. In this revised version and following the reviewer's suggestion, we have now counted the number of GFP electroporated cells and performed alternative normalization against the number of cells instead. The results show little variation but the reviewers advice was valid, the measurements show less variability when normalizing against the number of electroporated cells. They are more precise. The new quantifications provide an increase in statistic power when measuring S2 differences. Previous analysis revealed a tendency to decreases in S2 innervation in the shNrp1 condition. However, p values did not allow to conclude differences with control conditions. Normalizing against the number of GFP cells shows that the reductions in S2 innervation we observe upon knocking-down Nrp1 are significant. This sharpens our interpretations. We have changed the figures and text showing all data according to this method of normalization. We sincerely thank the reviewer for this suggestion.

We have developed a Macro for automatic counting that can be found as described in Material and Methods. We hope this is useful for other researchers.

3. On all graphs, the dots correspond to the sections analyzed and not the brains, so that the interindividual variability cannot be assessed. For all graphs and statistical analyses, the average of all sections for each brain should be used and not the value of each section. This is a critical point as it misleads the reader into assuming high power of analysis. At the very least three animals per condition (ideally at least 5) should be used to provide convincing conclusions. In addition, given the variability of GFP signal upon in utero electroporation, taking only 2 sections per brain seems very low. At least 3 to 4 sections should be quantified and averaged to avoid any bias.

The reviewer is right. We have corrected this mistake. In all figures, we now show the average obtained from the sections of each mouse as individual data points and then the mean and SEM value of all mice. Previously we showed the mean of all sections. Nevertheless, the data always included at least three mice per condition and stage. The number of animals necessary for conclusions is determined by statistical analysis. The number of animals per condition and the total number of mice in each analysis is now reflected in the figure legends.

4. For CTB injections, the injection sites for all brains should be shown (or a quantification of VPM / Po cell body location) in the supplementary figure.

We have included Figure 3 supplement 2 with all the injection sites.

5. For the CC analysis at the midline, is the dorso-ventral distribution similar all along the rostro-caudal axis? How is this parameter taken into account in the analyses? Are all electroporated sites equivalent in term of rostro-caudal position? This should be quantified, and the data should be normalized for this parameter.

In WT animals the dorso-ventral CC routes change some within the rostro-causal axis. However, opur quantifications are performed only at the level of the somatosensory area, where CC axons only follow ventral routes. Our electroporation protocol targets the SS cortex and our experiments and only include brains with equivalent size and location of electroporations at rostrocaudal and mediolateral positions. We have now included a new figure describing this Brains with obvious dissimilar electroporated areas are always discarded. Brains are selected visually after dissection under the dissecting scope and subsequently we confirm the correct location of the electroporated area by the analysis of the processed sections following histological landmarks (see Figure 4 supplement 1). We have included quantifications of the average distributions of electroporated cells in the radial axis, as well as in the mediolateral axis and in each section, and confirmed that the brains we analyzed in all groups are similar and not the source of our phenotypes (Figure 2 supplement 3 and Figure 4 supplement 1). There are minor changes in the mediolateral distribution of electroporated cells that were the source of minor changes in the dorsoventral path at the midline level. This is now corrected (see the response to reviewer#1).

Other concerns1. Timing of down / up regulation of Nrp1 expressionHere the shRNA against Nrp1 was electroporated at E15.5: is the down-regulation still efficient when callosal axons are extending and invading the contralateral cortex (ie. in the first postnatal weeks)? The levels of Nrp1 should be addressed in electroporated neurons, for example by RT-qPCR or western blot after FACS (this would allow quantifying differences of expression between S1 and S2 – see discussion l.290-291).

We electroporated the shRNA in plasmids. We and others have shown that expression of shRNAs pKLO lentiviral plasmids electroporated at E15.5 last to P21 postnatal stages (Cubelos et al., 2010; RodriguezTornos 2019), similar to the expression of CAG-GFP used as a reporter. See for example reduced Cux1 or Cux2 proteins in GFP-expressing layer 2/3 neurons of the P21 cortex electroporated at E15.5 with shRNA in supplementary materials in Cubelos et al., 2010. Unfortunately, detecting Nrp1 protein in the cell somas was not possible in our hands (manuscript pg 13 ln 323-325). We confirmed that the effects of shNrp1 in P16 brains are detectable after electroporation at E15.5 by assaying the effects of the constructs on the cingulate cortex, which express high levels of Nrp1. Whole tissue was dissected and Nrp1 levels assesed (Figure 2 supplement 1). Importantly, this comment has provided insightful revision. In addressing it, we have analyzed the patterns of Nrp1 throughout postnatal development and found patterns that help us to refine the interpretation of our data.

The neuropilin-semaphorin family was shown to play a role at the progenitor levels (Castellani lab). Since the constructs used here induce constitutive gain or loss of function, how can the authors be sure that the previous steps of development are not affected in L2/3 neurons?

As mentioned above, we have quantified the number and laminar distribution of GFP^+^ cells in all conditions and at all stages and we found no differences.

We now include analysis at earlier stages. This analysis at P10 demonstrates no differences between conditions. Importantly, it has revealed that the S2 column is formed after the S1 column and later than P10 in WTs and all our electroporating conditions. This finding, together with the description of nrp1 expression has allowed us to refine the interpretation of our data.

2. Role of Nrp1 in branchingThis is not directly addressed in the study and should be removed from the text. To formally conclude on this aspect, the authors should use sparse electroporation and morphological reconstruction of contralateral axons.

The role in branching is principally concluded from the comparative analysis of innervation patterns at different stages (current Figure 4 H-K). Our manipulations reduce contralateral GFP^+^ signal but not the number of CPNs. They block axonal increases that in controls are known to be caused by terminal branching. We have rewritten the text trying to make this clear and in the discussion specify that this is a model that requires further investigation.

3. Biological mechanism underlying the findingsNeuropilins do not act alone and need co-receptors of the plexin family. While this is an important point to understand the mechanisms, this is only superficially mentioned in the discussion (l. 317).

We now discuss this more extensively (pg 15 ln 378-392).

Reviewer #3 (Recommendations for the authors):The paper is a difficult read for non-specialists in the field. I suggest simplifying the language for the sake of broader audience. The final diagram might include postnatal refinement data.

We have rewritten most of the manuscript trying to address a broader audience. We have changed Figure 6 showing the diagram of the model.